# Evolutionary conserved NSL complex/BRD4 axis controls transcription activation via histone acetylation

Aline Gaub [1], Bilal N. Sheikh [1], M. Felicia Basilicata [1], Marie Vincent [2], Mathilde Nizon[2], Cindy Colson[3,4], Matthew J. Bird[5], James E. Bradner [6], Julien Thevenon[7], Michael Boutros[8,9] & Asifa Akhtar [1 ✉]

Cells rely on a diverse repertoire of genes for maintaining homeostasis, but the transcriptional networks underlying their expression remain poorly understood. The MOF acetyltransferase-containing Non-Specific Lethal (NSL) complex is a broad transcription regulator. It is essential in *Drosophila,* and haploinsufficiency of the human KANSL1 subunit results in the Koolen-de Vries syndrome. Here, we perform a genome-wide RNAi screen and identify the BET protein BRD4 as an evolutionary conserved co-factor of the NSL complex. Using *Drosophila* and mouse embryonic stem cells, we characterise a recruitment hierarchy, where NSL-deposited histone acetylation enables BRD4 recruitment for transcription of constitutively active genes. Transcriptome analyses in Koolen-de Vries patient-derived fibroblasts reveals perturbations with a cellular homeostasis signature that are evoked by the NSL complex/BRD4 axis. We propose that BRD4 represents a conserved bridge between the NSL complex and transcription activation, and provide a new perspective in the understanding of their functions in healthy and diseased states.

---

[1] Max Planck Institute of Immunobiology and Epigenetics, Stuebeweg 51, 79108 Freiburg, Germany. [2] CHU Nantes, Service de Génétique Médicale, 38 Boulevard Jean Monnet, 44000 Nantes, France. [3] Service Génétique, Génétique Clinique, CHU, Avenue Georges Clemenceau CS 30001, 14033 Caen, France. [4] Normandy University, UNICAEN, BIOTARGEN, Esplanade de la Paix CS 14032, 14032 Caen, France. [5] Department of Chronic Diseases, Metabolism and Ageing, Katholieke Universiteit Leuven, Herestraat 49, 3000 Leuven, Belgium. [6] Novartis Institutes for Biomedical Research, 181 Massachusetts Ave, Cambridge, MA 02139, USA. [7] CNRS UMR 5309, INSERM, U1209, Institute of Advanced Biosciences, Université Grenoble-Alpes CHU Grenoble, Allée des Alpes, 38700 La Tronche Grenoble, France. [8] Division of Signaling and Functional Genomics, German Cancer Research Center (DKFZ), Im Neuenheimer Feld 580, 69120 Heidelberg, Germany. [9] Department of Cell and Molecular Biology, Medical Faculty Mannheim, Heidelberg University, Theodor-Kutzer-Ufer 1-3, 68167 Mannheim, Germany. ✉email: akhtar@ie-freiburg.mpg.de

Chromatin modifications shape the gene expression patterns of a cell[1]. Loss of epigenetic regulators can lead to chromatin and transcriptional rewiring, and thus alter cellular identity and function. In humans, misregulation and mutation of epigenetic regulators have a significant pathological potential that can lead to e.g. developmental disorders or cancers[2,3]. While diagnosis of such diseases is greatly improving, for example due to the advances of next-generation sequencing[4], the relevant molecular understanding to establish a therapeutic approach is often missing. This remains particularly challenging for broadly expressed regulators, where the co-factors instructing their activity on ubiquitous versus tissue-specific target genes, remain often poorly defined. The MOF acetyltransferase-containing NSL complex is a major regulator of constitutively expressed genes in both flies and mammals. The NSL complex is essential for viability and conserved from flies to humans[5]. Heterozygous mutations in *KANSL1*, a subunit of the NSL complex, are causative for the Koolen-de Vries syndrome[6,7], which shows a prevalence of 1/16000, and is characterised by mild to moderate intellectual disability and co-morbidities including cardiac abnormalities and craniofacial defects[6–8]. Mutations in other subunits of the NSL complex including *MOF*[9] and *KANSL2*[10] are also associated with intellectual disability.

The spectrum of the NSL target genes encompasses diverse cellular functions from cell cycle to metabolism[11–14]. Tethering of the NSL complex to a reporter gene triggers strong transcription activation[15]. The NSL complex interacts with the NURF chromatin remodelling complex to ensure proper nucleosome positioning for accurate TSS selection[16]. However, a comprehensive picture of how the NSL complex, potentially with partner proteins, achieves transcription activation in flies or mammals is still missing.

The reduced molecular and genetic complexity of the *Drosophila* model system provides a powerful tool to interrogate regulatory networks of disease-relevant genes[17]. In the current study, we utilize *Drosophila* to generate a functional NSL complex interactome. We uncover an evolutionarily conserved functional interaction between the NSL complex and the BET protein BRD4 in transcription. Our work reveals an unexpected BET protein signature in patients with haploinsufficiency of *KANSL1*, which underlies the Koolen-de Vries intellectual disability syndrome.

## Results

**RNAi screen identifies functional NSL complex interactome.** The NSL complex is required for the proper expression of around 6000 genes in *Drosophila*[16], yet the mechanism by which it facilitates transcription remains unclear. We undertook an unbiased approach to identify the full functional interaction spectrum of the NSL complex in *Drosophila*. To this end, we performed a genome-wide RNAi screen based on a luciferase reporter assay in cultured cells, where tethering of NSL3, a subunit of the NSL complex, to a minimal promoter through a Gal4-DNA-binding domain conveys strong transcriptional activity[15] (Fig. 1a, Supplementary Fig. 1a). This assay was sensitive to depletion of other NSL complex members (Supplementary Fig. 1b), hence NSL1, NSL3 and MOF served as positive controls for the genome-wide screen. We used the *Drosophila* Heidelberg (HD2) library, which spans ~99% of the *Drosophila* protein-coding genome (14587 genes)[18]. Our assay allowed for detection of quantitive signal changes (Z′-factor = 0.73)[19] (Supplementary Figure 1c). Reassuringly, quality controls were fulfilled (Supplementary Fig. 1c–g) and we scored all seven NSL complex subunits in the genome-wide screen (Supplementary Fig. 1h), confirming its capability to detect functionally relevant factors. The screen provided us with both positive and negative regulators (Fig. 1b,

Supplementary Data 1), with a number of candidates displaying similar Z-scores as the depletion of MOF.

Based on Z-score thresholds, we selected the 367 top scoring genes for a secondary screen (Fig. 1b, c). In the secondary screen we performed two assays; first, we repeated the NSL3-driven reporter assay, which validated 80% of the rescreened candidates (Fig. 1d); second, we performed a control assay, where transcription activation is induced by the Gal4 activation domain (Gal4-AD) instead of NSL3. This additional assay enabled us to distinguish NSL complex-specific co-regulators from candidates generally involved in transcription and moreover, to control for Gal4 tethering as such. Among the factors that were scored in both NSL3 and Gal4-AD, we uncovered several components of the proteasome complex (Fig. 1e).

Several proteins having known functional relationships with the NSL complex were recovered amongst the screen targets. We identified the NURF complex (Z-score < −6 for Caf1, E(bx), Iswi) (Fig. 1f) as a functional modulator of NSL3-driven transcription. This is consistent with our previous work demonstrating that functional and physical interactions between the NSL and NURF complexes are essential for faithful nucleosome positioning at NSL complex-bound promoters in *Drosophila*[16]. Furthermore, we also scored Chromator (Chro) and Putzig (Pzg), which have both been reported to co-purify with the NSL complex[5,15] and could possibly relate to a role in genome organization[20,21].

The vast majority of targets uncovered by the screen were novel factors without any previous functional associations with the NSL complex. For example, we scored multiple subunits of different chromatin-associated complexes, such as the RNA Polymerase 2 Associated Factor 1 (PAF1) (Z-score < −6 for Atms, Hyx), TIP60 (Z-score < −6 for Dom, E(pc), Act87E, Bap55, His2Av) and the Myb/dREAM complex (Z-score < −6 for Caf1, Dp, Mip120, Mip130) (Fig. 1f). Furthermore, components of the SCF-slmb complex (Z-score < −6 for all components) and the PP2A complex (Z-score < −6 for Mts, Pp2A-29B, Pp2A-B′, Tws) were scored. These complexes harbour enzymes responsible for protein ubiquitination and dephosphorylation, respectively[22,23]. Using the Search Tool for the Retrieval of Interacting Genes/Proteins (STRING) database of physical and functional interactions[24], we found that the targets identified from the screen show significantly higher network connectivity than expected by random chance (*p*-value < 1.0e-16) (Supplementary Fig. 2). Moreover, we identified categories such as hippo signalling, hedgehog signalling, spliceosome and ubiquitin mediated proteolysis as overrepresented.

One of the highest scoring candidates in our screen was dBRD4/fs(1)h (Fig. 1b, e, f), the only Bromodomain and Extra Terminal Domain (BET) family protein in *Drosophila*[25]. dBRD4, like *BRD4* in mammals, produces two distinct isoforms (long isoform dBRD4-L and short isoform dBRD4-S), that both associate with chromatin[25]. BRD4 promotes productive elongation by recruiting P-TEFb, which phosphorylates RNA Polymerase 2 (Pol2)[26]. Furthermore, BRD4 has been shown to associate with multiple developmental transcription factors such as Myc, IRF4 and PRDM1 in health and disease[27–29]. Given that the NSL complex predominantly regulates constitutively expressed genes, we were curious to understand how it could be functionally related to an established regulator of inducible gene expression such as BRD4.

**The NSL complex and dBRD4 colocalize on endogenous promoters.** We observed a strong (>75%) and reproducible decrease of NSL3-driven luciferase reporter expression following dBRD4 RNAi in our pilot, genome-wide and secondary screens (Fig. 2a). To validate the functional interaction between the NSL complex

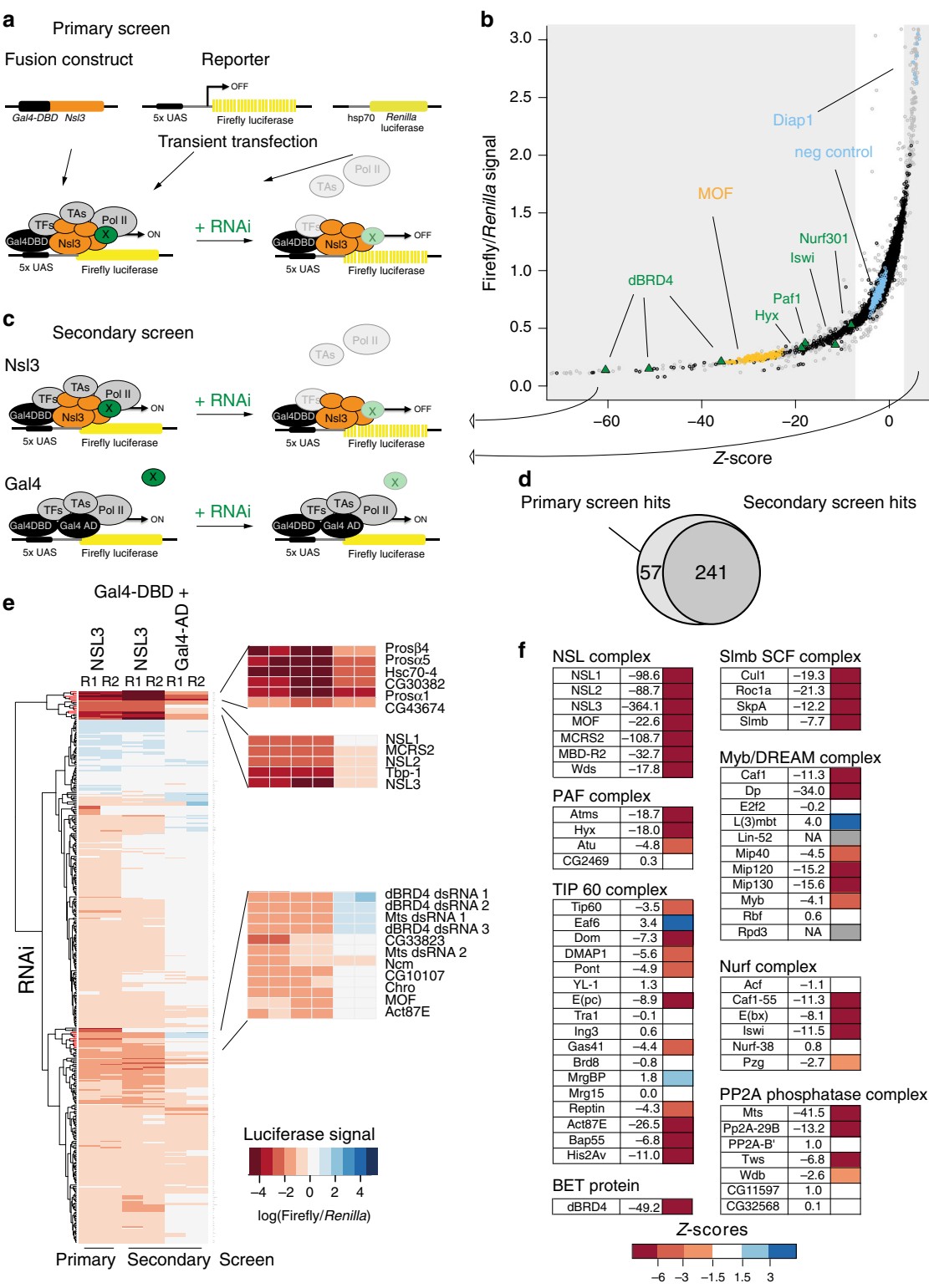

and dBRD4 in an endogenous context, we examined their genome-wide distributions in S2 cells. A comparison between publicly available dBRD4[25] and NSL3[13] ChIP-seq binding profiles showed that a remarkable 86% of NSL3-bound promoters were co-occupied by dBRD4 (calculation based on MACS peaks, p-value < 2.2e-16) (Fig. 2b, c). This extensive co-localization of NSL3 and dBRD4 on chromatin was further confirmed by immunofluorescence staining of polytene chromosomes in dBRD4-L expressing transgenic flies (Fig. 2d). Gene Ontology

(GO) enrichment analysis on gene promoters co-occupied by both NSL3 and dBRD4 revealed biological processes such as Cellular Metabolism, RNA processing, Intracellular Transport and Biosynthesis (Fig. 2e). Many of these overlap with GO terms associated with NSL complex targets[11–13,30]. This analysis therefore uncovered two unexpected findings. First, that the majority of NSL complex and dBRD4 genome-wide gene targets are common. Second, that dBRD4 displays binding to genes responsible for cellular and metabolic homeostasis, suggesting

**Fig. 1 RNAi screen identifies functional NSL complex interactome. a** Scheme of primary genome-wide screen. Plasmids containing a fusion construct, *nsl3* gene fused to Gal4 DNA-binding domain (*Gal4-DBD*), a firefly *luciferase* reporter containing Gal4 DNA-binding upstream activating sequence elements (5xUAS) and a constitutively active hsp70 Renilla *luciferase* reporter were co-transfected into S2 cells. Effect of RNAi of a candidate (X) on reporter signal is assayed. Renilla signal serves a control for transfection efficiency. **b** Data distribution of primary RNAi screen. Scatterplot of Z-scores and luciferase signal (average of two replicates) are plotted for each knockdown (within the Z-score range of −70 to +6). Data points from grey shaded areas were used for secondary screen. Grey datapoints: candidates excluded due to strong effect on Renilla signal (see Methods for more details on filtering and analysis of RNAi screen), orange datapoints: positive control knockdowns, blue datapoints: negative control knockdowns (GST, GFP and Diap1), green triangles: other candidates (dBRD4, Nurf complex and PAF complex). **c** Scheme of secondary screen reporter assay. Upper part as in **a**, lower part: Gal4-DBD fused to Gal4 activation domain (*Gal4-AD*) represents the canonical full length *Gal4*. Full length *Gal4* it is used as control, to discriminate NSL unspecific transcription factors. **d** Venn diagram depicting overlap of candidates that scored in the primary and secondary *Nsl3-Gal4-DBD* screens. The same thresholds for firefly luciferase signal relative to Renilla luciferase signal were applied for both primary and secondary screens. **e** Heatmap of log-scaled fold changes of normalized luciferase signal in the primary and secondary RNAi screens. Results for the 367 knockdowns performed in the secondary assays are plotted. The order of genes was generated by unsupervised hierarchical clustering. **f** Z-scores of genome-wide RNAi screen for several complexes and protein categories are listed. If a gene was targeted by multiple dsRNAs, an average of the respective Z-scores is given.

that dBRD4 has a far more significant contribution to their regulation than was previously anticipated.

**dBRD4 promotes transcription elongation of NSL target genes.** Given the significant overlap between dBRD4 and NSL complex target genes, we wanted to determine the relevance of their shared occupancy for transcription. A number of small molecule inhibitors of BET proteins, including JQ1 and iBET are available[31,32]. Furthermore, thalidomide-conjugated JQ1 molecules (dBET) that permit rapid degradation of BET proteins in mammalian cells have recently been generated[33]. JQ1 has been shown to be functional in *Drosophila*[25]. We observed that treating *Drosophila* cells with dBET led to the complete degradation of dBRD4 within 4 h (Fig. 3a). These small molecule inhibitors thus provide an opportunity to study the effect of acute loss of dBRD4 function independent of the secondary effects that can arise from RNAi treatment. We detected a substantial decrease in NSL3-driven reporter gene expression in cells treated with JQ1, iBET or dBET, with stronger effects observed upon longer treatment (16 h vs. 9 h) (Fig. 3b–d, Supplementary Fig. 3a–c). Conversely, the Gal4-driven control reporter activity rather increased (Fig. 3b–d, Supplementary Fig. 3a–c). This indicates that the transcriptional effects of dBRD4 loss are specifically mediated through functional crosstalk with the NSL complex and do not arise as a result of a global expression defect. We observed a similar response when tethering the human orthologue of NSL3, KANSL3, to the promoter of the luciferase reporter in *Drosophila* cells, suggesting that the transcriptional link between dBRD4 and the NSL complex is evolutionarily conserved (Fig. 3b–d, Supplementary Fig. 3a–c).

Having shown that depletion or inhibition of dBRD4 reduces NSL3-dependent transcription of a luciferase reporter, we next probed the effects of dBRD4 loss on endogenous NSL target genes. We undertook total RNA-seq experiments in S2 cells after NSL1, NSL3, dBRD4 and control (GST) RNAi as well as 1 h JQ1, 4 h JQ1 and DMSO treatment (Fig. 3e). Depletion of NSL complex members and JQ1 treatment affected the expression of thousands of genes (Fig. 3e, Supplementary Fig. 3d, e), and consequently standard RNA-seq normalization methods could not be applied. Instead, we normalized the RNA-seq data to External RNA Controls Consortium (ERCC) spikes that were added prior to library generation. Using this analysis, we scored 7362 differentially expressed (DE) genes for NSL1 RNAi, 5995 DE genes for NSL3 RNAi, 3503 DE genes for dBRD4 RNAi, 3835 DE genes for 1 h JQ1 and 6609 DE genes for 4 h JQ1 (false discovery rate (FDR) < 0.05) (Supplementary Data 2). Consistent with its role as a transcriptional activator, NSL complex-bound targets were downregulated upon NSL1 RNAi ($p$-value = $1.2e^{-7}$) (Fig. 3f,

Supplementary Fig. 3f). As expected, there was significant correlation between DE genes scored upon NSL1 and NSL3 RNAi (adj $R^2 = 0.56$). We validated the RNA-seq results using RT-qPCR measurements (Supplementary Fig. 3g, h). In agreement with previous findings[25], we detected a smaller number of differentially expressed genes upon dBRD4 RNAi compared to dBRD4 inhibition (Supplementary Fig. 3g, h). This may be a consequence of adaptation of the cells during the four days of RNAi.

From the global RNA-seq experiments, we uncovered a significant correlation between gene expression changes observed upon depletion of NSL1 and inhibition of dBRD4 by JQ1 (4 h) (adj $R^2 = 0.20$, Fig. 3g). In fact, 84% of the genes that were differentially downregulated upon NSL1 RNAi were also differentially downregulated following 4 h of JQ1 treatment (Fig. 3e). Likewise, NSL complex-bound genes were significantly downregulated after knockdown of either the long isoform or both isoforms of dBRD4 using RNAi (Fig. 3f, Supplementary Fig. 3f), whereas genes not bound by the NSL complex were not significantly affected (Supplementary Fig. 3f). These correlations suggest that dBRD4 and the NSL complex not only co-occupy the same target genes but also cooperate in enabling their transcription.

As an important control, we checked the protein levels of NSL complex members after dBRD4 RNAi. We did not observe any change in the stability of several NSL complex components after dBRD4 depletion. Conversely, dBRD4 protein levels also did not change upon NSL1, NSL3 or MOF RNAi (Supplementary Fig. 3i). This supports the idea that dBRD4 affects NSL target gene expression through transcriptional interplay with NSL complex members and not by affecting NSL protein levels.

We were interested in pinpointing which steps of transcription were influenced by dBRD4 inhibition. We had previously reported that depletion of the NSL complex affects the recruitment of RNA Pol2 to promoters[13]. In line with previous findings, cells depleted of NSL1 showed a bulk decrease in both Pol2 serine-5 phosphorylation (Pol2 ser5p), which is associated with transcription initiation, and Pol2 serine-2 phosphorylation (Pol2 ser2p), which is associated with transcription elongation (Fig. 3h). In contrast, treating cells with JQ1 or dBET only led to a decrease in Pol2 ser2p, while Pol2 ser5p remained largely unaffected (Fig. 3i, Supplementary Fig. 3j). To determine, whether these bulk-level effects were reflected in transcriptional changes at NSL target genes, we assessed RNA Pol2 occupancy by Rpb3 ChIP-seq in cells treated with JQ1 or dBET (Fig. 3j). Inhibition or degradation of dBRD4 led to a moderate increase of total Pol2 at promoters and a decrease in occupancy throughout the gene bodies. These effects were more pronounced for NSL-bound genes compared to expressed, NSL-non-bound genes (Fig. 3j). In

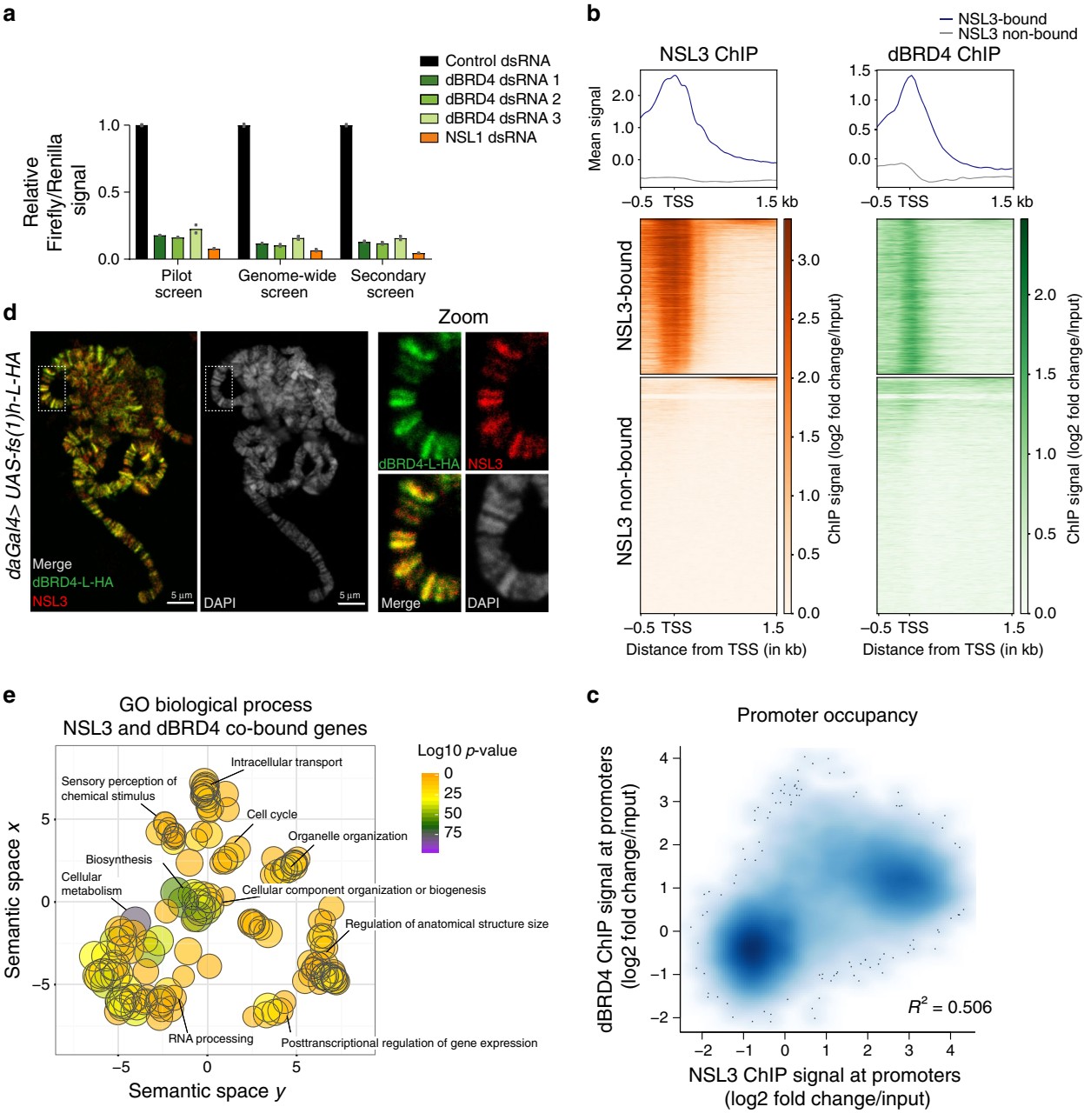

**Fig. 2 The NSL complex and dBRD4 colocalize on endogenous promoters. a** Barplot of mean firefly signal normalized to Renilla signal and control RNAi (GST) of each knockdown and screen (two replicates). Source data are provided as a Source Data file. **b** Heatmaps and summary plots of NSL3[13] and dBRD4[25] ChIP profiles. Log2 fold changes over input are plotted for all genes. Genes of both heatmaps are clustered and sorted based on NSL3 ChIP signal. $p$-value < 2.2e-16 for one-sided Fisher's Exact Test on overrepresentation of dBRD4 binding on NSL3-bound promoters. **c** Scatterplot of NSL3[13] and dBRD4[25] ChIP signals on gene promoters (TSS ± 200bp). Log2 fold changes over input are plotted for all gene promoters. $R^2$ was calculated with linear regression model. **d** Polytene chromosome immunostaining of *daGal4 > UAS-fs(1)h-L-HA* third instar larvae. NSL3 staining in red, HA staining in green: tagged long isoform of dBRD4 (dBRD4-L-HA). Scale bar: 5 μm. This experiment was repeated independently two more times showing similar results. **e** Gene Ontology (GO) Enrichment analysis of Biological Processes of genes promoter-bound by both dBRD4 and NSL3. Promoter-bound genes were defined by MACS peak calling on dBRD4[25] and NSL3[13] ChIP-seq datasets considering TSS ± 200 bp as promoter region. Enriched GO terms were visualized with REVIGO.

contrast, depletion of NSL1 affected both the promoters and gene bodies of NSL-bound genes. These changes are unlikely to be a consequence of ChIP efficiency or normalization, as the spiked-in *Drosophila virilis* chromatin was equally recovered in controls and inhibitor-treated cells (see Methods). We conclude that RNA Pol2 elongation is defective at NSL-bound genes when dBRD4 is inhibited or degraded. Our data furthermore suggest, that the NSL complex and dBRD4 collaborate in a functional relay coupling Pol2 transcription initiation and elongation events on NSL target genes.

**Efficient recruitment of dBRD4 requires the NSL complex.** The transcriptional effects observed upon loss of dBRD4 and the NSL complex implied that dBRD4 recruitment requires the NSL complex. To test this hypothesis, we performed dBRD4 ChIP-seq experiments in NSL1 RNAi-treated cells. We generated a stable

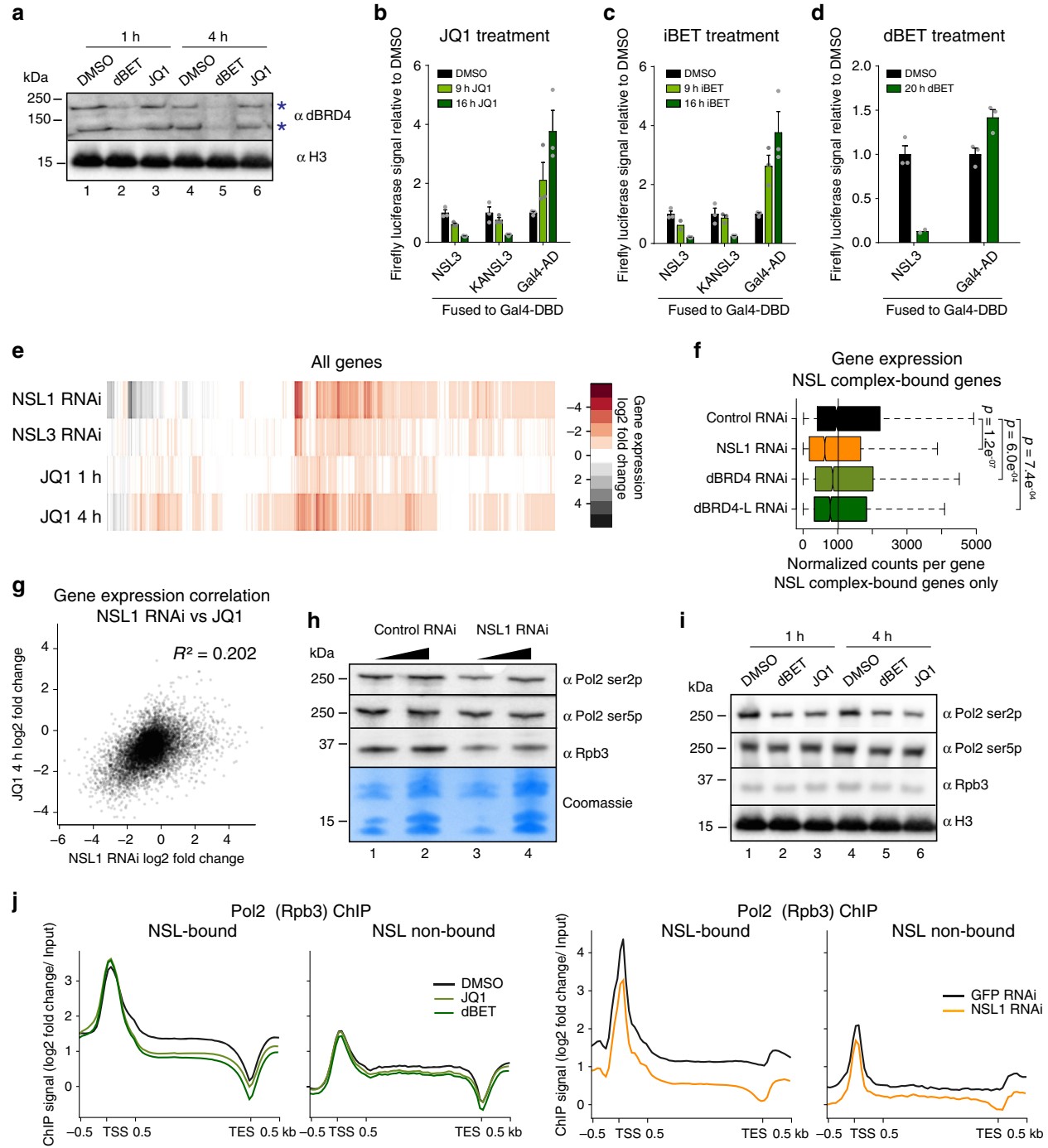

cell line expressing the FLAG-His-Bio-His tagged short isoform of dBRD4 (dBRD4-S-Biotin)[34] under the control of an inducible MtnA promoter. This enabled us to use stringent wash conditions and untagged controls, while inducing the tagged protein at similar levels across different conditions (Supplementary Fig. 4a, b). We used exogenous biotinylated DNA spikes to control for ChIP efficiencies in the different RNAi conditions (see Methods). We performed peak calling for both the dBRD4-S-Biotin and the dBRD4 antibody ChIP data[25]. We identified 2491 dBRD4-S-Biotin-bound promoters (Supplementary Fig. 4c, d), which represents a large subset of the 4525 dBRD4 antibody ChIP-associated promoters. The biotin ChIP-seq experiments revealed a robust decrease of dBRD4 occupancy following NSL1 and MOF RNAi (Fig. 4a) at the majority of dBRD4-S-Biotin-bound promoters, while binding at few sites remained unaffected

(Supplementary Fig. 4e). Changes in dBRD4 occupancy were most striking for NSL1 RNAi followed by MOF RNAi (Fig. 4a–c; qPCR validation in Fig. 4d).

To determine whether dBRD4 depletion in turn affects NSL complex occupancy, we performed the reciprocal ChIP-qPCR experiments for NSL1 and MOF after JQ1 treatment, dBET treatment or dBRD4 RNAi. We assessed their enrichment at promoters of genes that showed decreased expression levels upon both NSL1 RNAi and JQ1 treatment (Supplementary Fig. 4f). There was no decrease of NSL1 or MOF occupancy at any of the tested genes after treatment with JQ1 or dBET or after dBRD4 RNAi (Fig. 4e–g, Supplementary Fig. 4g, h). Together with the dBRD4 ChIP analyses, these results suggest that the NSL complex functions upstream of dBRD4. While the NSL complex is required to recruit dBRD4 to target gene promoters, both the

**Fig. 3 dBRD4 promotes transcription elongation of NSL target genes. a** Western blot for dBRD4 and H3 after 100 nM dBET6 (lanes 2 and 5) or 5 μM JQ1 (lanes 3 and 6) treatment. Blue asterisks indicate dBRD4-S and dBRD4-L. The experiment was repeated twice showing similar results. **b–d** Firefly luciferase activity using NSL3 and KANSL3 (human orthologue of NSL3) fused to Gal4 DNA-binding domain or full-length Gal4 to drive expression of the *UAS-firefly luciferase* reporter upon (**b**) 5 μM JQ1, (**c**) 1 μM iBET 762 or (**d**) 100 nM dBET6 treatment. Bars represent mean values ± SEM (*n* = 3 technical replicates). **e** Heatmap of total RNA-seq. Log2 fold changes of gene expression in NSL1 and NSL3 RNAi and JQ1 (5 μM) treatments for 1 h or 4 h versus control RNAi (GST) or DMSO for all expressed genes are plotted. Gene order was generated by unsupervised hierarchical clustering. **f** Boxplot of normalized RNA-seq counts in NSL1, dBRD4, dBRD4-L and control RNAi (GST) for NSL complex-bound genes (*n* = 5600). Two-sided Welch two sample *t*-test was applied. Boxplots show median (centre), interquartile-range (box) and minima/maxima (whiskers). **g** Scatterplot of gene expression changes after 4 h JQ1 treatment and NSL1 RNAi. Log2 fold changes for all genes are plotted. Linear regression model was applied. **h** Representative western blot for Pol2 ser2p, Pol2 ser5p and Rpb3 after NSL1 RNAi (lanes 3 and 4), for quantification see Supplementary Fig. 3j. H4 blot shown here is identical to **a**. **i** Representative western blot for Pol2 ser2p, Pol2 ser5p, Rpb3 and H3 after 100 nM dBET6 (lanes 2 and 5) or 5 μM JQ1 (lanes 3 and 6) treatment, for quantification see Supplementary Figure 3j. **j**. Average profiles of Pol2 (Rpb3) ChIP-seq signal for expressed NSL-bound (*n* = 5600) and expressed non-NSL-bound (*n* = 1600) genes after 1 h JQ1 (5 μM), 1 h dBET6 (100 nM) or NSL1 RNAi[13] compared to controls (DMSO and GFP RNAi). Gene bodies are scaled from 0.5 kb until TESs. *Drosophila virilis* chromatin was added to experimental *Drosophila melanogaster* chromatin before Rpb3 ChIPs to control for IP efficiency and library composition effects (see Methods). **e–g** Expression was normalized using synthetic ERCC spikes (see Methods). *n* = 3 biological replicates. Source data for **b–d** are provided as a Source Data file.

NSL complex and dBRD4 are necessary for productive transcription. This is confirmed by our observation that gene expression is abolished in the absence of dBRD4, even if the NSL complex is still bound at the respective gene promoters (Fig. 3).

**NSL complex-mediated acetylation leads to dBRD4 recruitment.** The NSL complex harbors histone acetyltransferase activity in the form of its enzymatic member MOF. In the context of the NSL complex, MOF not only catalyzes the acetylation of H4K16, but also of H4K5, K8 and K12 in vitro[30,35]. As BET proteins are capable of recognizing acetylated histone residues[36,37], we wanted to investigate whether this mark represents the link between the NSL complex and dBRD4. While the affinity of bromodomains towards mono acetylated histones is modest, it increases with the presence of multiple acetylated residues in tandem[36,38,39]. We therefore tested whether the different H4 tail modifications, which can be catalyzed by MOF, would directly impact on the binding affinity of dBRD4 to the H4 tail in vitro.

To this end, we performed biolayer interferometry measurements. We expressed dBRD4-S in bacteria and quantified its binding to N-terminal histone H4 peptides harboring different acetylation patterns (see Methods for details). As controls, we used bacteria expressing the histone H3 lysine 9 methylation-specific reader HP1[SWI6], as well as uninduced bacteria that do not express any recombinant protein (Supplementary Fig. 5a). The assay revealed a strong binding of dBRD4-S to the multi-acetylated H4K5,8,12,16ac peptide and an intermediate binding affinity to the di-acetylated H4K12,16ac peptide. The binding to the mono-acetylated peptides H4K12ac and H4K16ac was weaker and comparable to the background signal of the uninduced control (Supplementary Fig. 5b, c), which is consistent with previously published data on bromodomain binding affinities[36]. Furthermore, these binding events occurred in a dBRD4-dependent fashion, since lowering its concentration by dilution with uninduced extracts led to a concordant reduction of the detected binding signal (Supplementary Fig. 5c). Importantly, HP1[SWI6] did not display binding to any of the acetylated histone H4 peptides tested (Supplementary Fig. 5c), which confirms the specificity of our experimental setup. These results demonstrate that NSL-mediated H4 tail modifications, including H4K16ac can be recognized by dBRD4 and as for other bromodomain proteins, the presence of multiple acetylations promotes its binding to the histone tail.

To investigate whether NSL deposited histone acetylation provides the connection to dBRD4 recruitment in vivo, we performed H4K16ac and H3 ChIP-seq experiments in control and NSL1 RNAi cells. We used H4K16ac as a proxy for NSL-mediated histone acetylation, as changes in H4K16ac can be

directly linked to MOF, while H4K5, K8 or K12 acetylations can also be catalyzed by other HATs[12,40,41]. We compared NSL-bound and non-bound genes. NSL-bound genes displayed strong changes in dBRD4-S-Biotin and Pol2 ChIP signal (Fig. 5a). Similarly, we found that the NSL-bound group also displayed reduced H4K16ac levels upon NSL1 RNAi (Fig. 5a). Furthermore, overall H4K16ac levels at promoters positively correlated with dBRD4 binding (adj $R^2$ = 0.25, Supplementary Fig. 5d). These data suggest that NSL complex-mediated acetylation and dBRD4 recruitment may indeed be linked in vivo.

We designed a rescue experiment to test the requirement of NSL complex-mediated histone acetylation for recruitment of dBRD4 to target promoters. Treating cells with the histone deacetylase inhibitor (HDACi) panobinostat increased overall histone acetylation levels, including H4K16ac, in both control and NSL1 RNAi cells (Supplementary Fig. 5e). This was also reflected in ChIP-qPCR, where H4K16ac increased on both promoters and gene ends (Fig. 5b, right panel). Remarkably, we found that HDACi treatment rescued dBRD4 recruitment in NSL1 RNAi cells (Fig. 5b, left panel), although the extent of rescue varied between targets. In contrast, NSL1 and MOF occupancy remained entirely unaffected by HDACi treatment (Supplementary Fig. 5f, g). These results demonstrate that a change in acetylation status is sufficient to elicit recruitment of dBRD4 even in the absence of the NSL complex.

Together this data suggests that NSL-mediated histone H4 acetylation impacts on chromatin binding of dBRD4 and establishes a hierarchical NSL/BRD4 axis for transcription activation.

**The NSL complex/BRD4 axis is conserved in mice and humans.** Having identified the important role played by the NSL/BRD4 axis in transcription activation of *Drosophila* genes, we wondered whether it might also be conserved in mammals. Indeed, we had observed earlier that BET inhibitors impaired luciferase reporter activation by a tethered human KANSL3 (Fig. 3b, c). Therefore, we performed comparative analyses of NSL complex and BRD4 binding profiles in mouse embryonic stem cells (mESC) using available ChIP-seq datasets[11,42,43]. BRD4 showed strong binding to NSL-occupied promoters in mESCs (Supplementary Fig. 6a). In addition, BRD4 binding patterns appeared overall very similar between three different cell types of different developmental stages: ESCs, Embryoid Bodies (EB) and mixed lineage leukemia (MLL) cells (Supplementary Fig. 6a). Such a pattern would be expected for a factor regulating constitutive gene expression and could hint towards a co-regulation with the NSL complex[11–13].

To explore this further, we performed expression analysis in mESCs upon depletion of BRD4 and KANSL2. While *Drosophila*

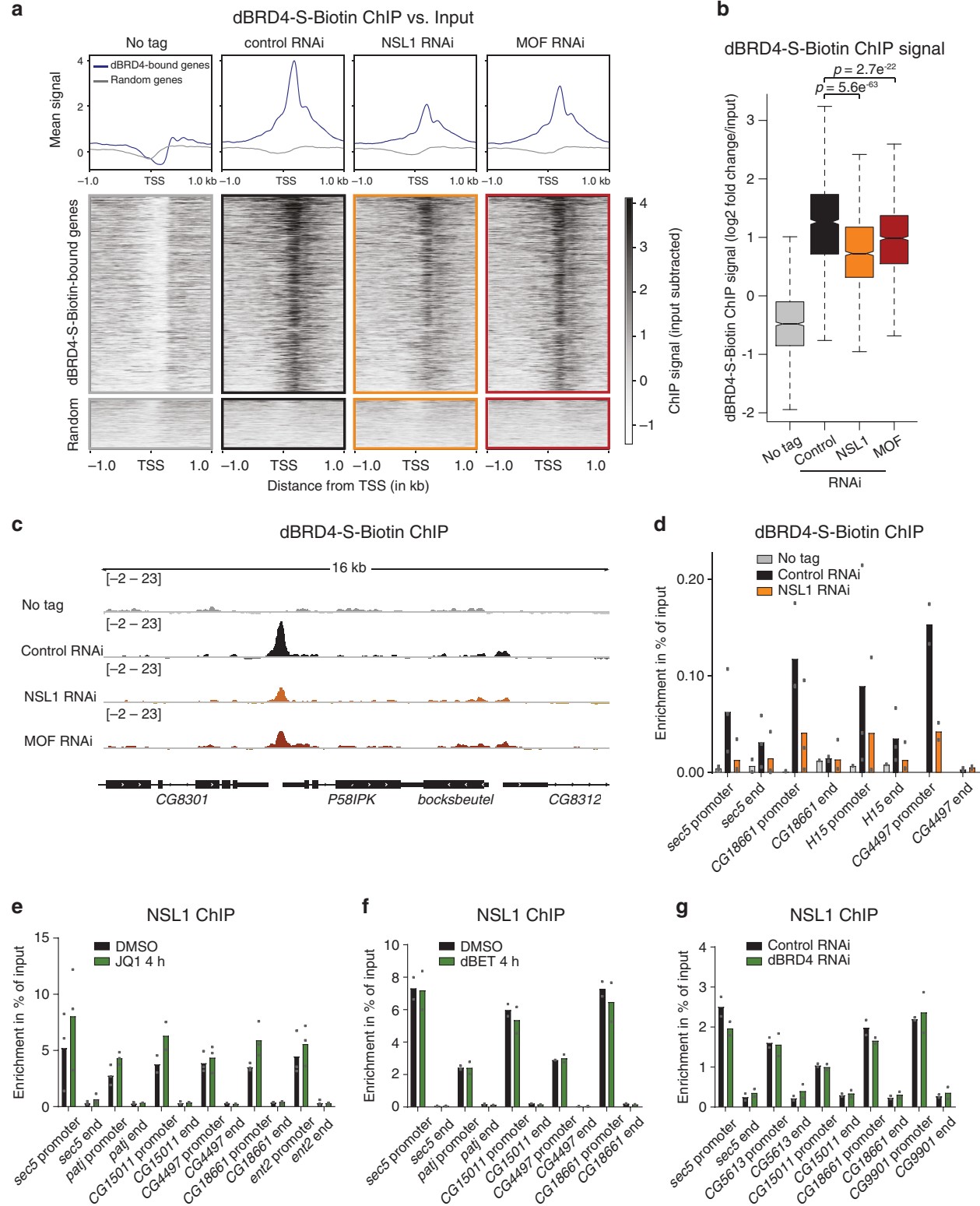

possess just one BET protein (dBRD4), there are three BET homologues in mice and humans: BRD2, BRD3 and BRD4[25]. As the small molecule dBET induces degradation of all BET proteins, dBET treatment in mammalian cells approximates the loss of dBRD4 in *Drosophila*. In turn, we generated *CAG-Cre-ERT2 Kansl2*[fl/fl] mESCs for conditional knockout (KO) of *Kansl2* (Supplementary Fig. 6b). As the whole NSL complex is required

for transcription activation (Fig. 1f), loss of KANSL2 is expected to exemplify NSL complex function more generally. We used 2i media for mESC culture to ensure a naïve state of the cells throughout the experiments. We compared gene expression changes by RT-qPCR and observed that different from dBET, pluripotency (e.g. *Nanog*) and differentiation (e.g. *Cdh2*) markers were unaltered upon *Kansl2* deletion (Supplementary Fig. 6c).

**Fig. 4 Efficient recruitment of dBRD4 requires the NSL complex.** Experiments addressing dBRD4 targeting were performed in a stable S2 cell line, where the short isoform of dBRD4 triple tagged with FLAG, His and biotin (dBRD4-S-Biotin) was under the control of a copper-inducible promoter. For all experiments with this stable cell line, expression of dBRD4-S-Biotin was induced by addition of 1 mM $CuSO_4$ to the medium for 16 h. **a** Heatmaps and profile plots of dBRD4-S-Biotin ChIP-seq in control (GST RNAi), NSL1 RNAi, MOF RNAi and WT untagged cells. The upper cluster comprises 2491 dBRD4-S-Biotin-bound and endogenous dBRD4-bound genes, the lower cluster of a similar number of randomly selected, not dBRD4-S-Biotin-bound genes. Both clusters are ordered by average intensity of all four ChIP datasets. Input-subtracted ChIP-seq data (merge of two biological replicates) is shown. **b** Boxplot of dBRD4-S-Biotin ChIP signal (log2 fold change over input) (merge of two biological replicates) on promoters of dBRD4-S-Biotin-bound genes ($n = 2491$) (±200bp from TSS) for the respective RNAi experiment. Boxplots show median (centre), interquartile-range (box) and minima/maxima (whiskers). Two-sided Welch two sample $t$-test was applied. **c** Representative IGV browser snapshots of dBRD4-S-Biotin ChIP-seq profiles in control (GST), NSL1 and MOF RNAi as well as in WT untagged cells, input-subtracted ChIP-seq data (merge of two biological replicates) is shown. **d** dBRD4-S-Biotin ChIP-qPCR in control (GST) and NSL1 RNAi as well as in WT cells serving as untagged control. Primers target the promoters and ends of dBRD4-bound genes. **e** NSL1 ChIP-qPCR of cells treated with JQ1 (5 µM) or DMSO for 4 h. **f** NSL1 ChIP-qPCR of cells treated with dBET6 (100 nM) or DMSO for 4 h. **g** NSL1 ChIP-qPCR of cells after control (GST) or dBRD4 RNAi treatment. For **d–g** data are presented as mean values, for **d**, **e** $n = 2$ or 3 biological replicates, for **f**, **g** $n = 2$ biological replicates. **e–g** Primers target the promoters and ends of NSL complex and dBRD4-bound genes, for expression changes in 4 h JQ1 treatment of these genes, see Supplementary Fig. 4f. Source data for **d–g** are provided as a Source Data file.

However, concordant gene expression changes on NSL-target genes were detected in both *Kansl2* deletion or dBET-treated cells (Fig. 6a, Supplementary Fig. 6d).

To test whether the gene expression changes are associated with a loss of targeting of BET proteins, we performed BRD4 ChIP-qPCR experiments in *Kansl2* KO and dBET-treated cells. ChIP-qPCR from dBET-treated cells confirmed the specificity of the obtained BRD4 ChIP signal. Remarkably, targeting of BRD4 was impaired in the absence of the NSL complex member KANSL2 (Fig. 6b, Supplementary Fig. 6e).

Since pluripotency is unaffected in this scenario (Supplementary Fig. 6c), loss of BRD4 occupancy appears to be a specific consequence of impaired NSL complex function, rather than through a change in cellular state. The results additionally suggest that the functional interaction and recruitment hierarchy between the NSL complex and BRD4 that we characterized in *Drosophila* cells is evolutionary conserved in mammalian cells.

The Koolen-de Vries syndrome is a complex neurodevelopmental disorder caused by haploinsufficiency of *KANSL1*[6,7]. Given the high level of evolutionary conservation in NSL complex targets between flies and mammals[14], we hypothesized that the BRD4-related gene expression signature would also be evident in affected individuals. To address this question, we cultured primary fibroblast cell lines from Koolen-de Vries patients and healthy controls and performed RNA-seq experiments. Despite the sizeable heterogeneity between patient samples, we could detect 198 DE genes (FDR < 0.05, Fig. 6c, Supplementary Data 2, Supplementary Fig. 7a, b). When looking at directionality of expression change, we found that genes whose orthologues are bound by KANSL3 in mESCs had a significantly higher probability of being downregulated in patient cells ($p$-value = 4.8e$^{-6}$). Moreover, we identified a set of conserved target genes, which were consistently downregulated in *Drosophila* upon NSL1 RNAi (FDR < 0.05), in mice upon *Kansl3* knockdown (FDR < 0.05,[11]) and in *KANSL1*-haploinsufficient Koolen-de Vries patient cells (FDR < 0.4) (Fig. 6d, Supplementary Fig. 7a, b). The majority of those were direct NSL targets in *Drosophila* and mESCs by ChIP and comprised mostly genes involved in metabolic processes, such as mitochondrial function (e.g. *MICU2*) and lipid metabolism (e.g. *ABHD3*). These results suggest that the transcriptional role of the NSL complex contributes to expression deregulation in *KANSL1* haploinsufficient patients. We next investigated the gene expression signature specific to these patient cells more closely. Pathway analysis revealed broad categories such as Organismal Injuries and Abnormalities and Cancer as most significantly affected disease and function categories. In addition, the gene groups Cellular Movement, Cellular Development and Cellular Growth and Proliferation were among the top

10 most deregulated categories (Fig. 6e). Consistent with the housekeeping gene regulatory function of the NSL complex, these pathway analyses likely reflect an overall imbalance of cellular and metabolic homeostasis in *KANSL1* haploinsufficient patients.

We then compared Koolen-de Vries syndrome expression profiles to JQ1-treated MOLT4 cells[33] (also see Supplementary Fig. 3e). We chose this JQ1 dataset, because it contained ERCC spikes for normalization. We classified genes based on their response to JQ1 into JQ1-up, JQ1-down or JQ1-not DE. We then looked at the response in these three groups of genes in the Koolen-de Vries patient fibroblast dataset. We observed that the JQ1-down group is predictive of a downregulation in the patient cells ($p$-value = 1.5e$^{-49}$ vs. not DE group, Fig. 6f, Supplementary Fig. 7c). This overlap of BET and KANSL signatures was also evident when we classified transcriptome alterations in patients according to directionality of the change (up- or downregulation). KANSL3-bound genes ($p$-value = 4.8e$^{-06}$) and JQ1-responsive genes ($p$-value < 2.2e$^{-16}$), were significantly overrepresented in the downregulated group (Fig. 6g), whereas genes bound by MSL2, a subunit of another MOF-associated complex, were not overrepresented.

Our findings strongly suggest that the functional interaction between the NSL complex and BRD4 is evolutionarily conserved and critical for robust expression of their shared target genes. Furthermore, we reveal that the NSL complex/BRD4 axis might represent a clinically relevant and potentially druggable aspect of NSL complex-associated diseases in humans.

## Discussion

In this study, we took advantage of the reduced genetic complexity of the *Drosophila* system to screen for functional interaction partners of the NSL complex. Improving our understanding of transcription of NSL target genes is imperative given that haploinsufficiency of *KANSL1* is causative of the debilitating Koolen-de Vries syndrome. Employing a genome-wide approach, we revealed the diverse functional interaction network of the NSL complex in *Drosophila*. This screen permitted us to identify BRD4 as a critical co-factor for transcriptional activation at NSL target genes (see model Fig. 7). Herewith, we uncover a previously underappreciated role of BRD4 in regulation of thousands of constitutively expressed genes, including genes involved in cellular metabolism, RNA processing and cell cycle. We show that BRD4 and the NSL complex share a common set of target genes, and reveal that functional cooperation between the NSL complex and BRD4 is necessary for transcription of these. Mechanistically, the NSL complex and BRD4 appear to participate in a functional relay, which guides RNA Pol2 through the initiation-to-elongation transition. We show that NSL complex binding mediates deposition of

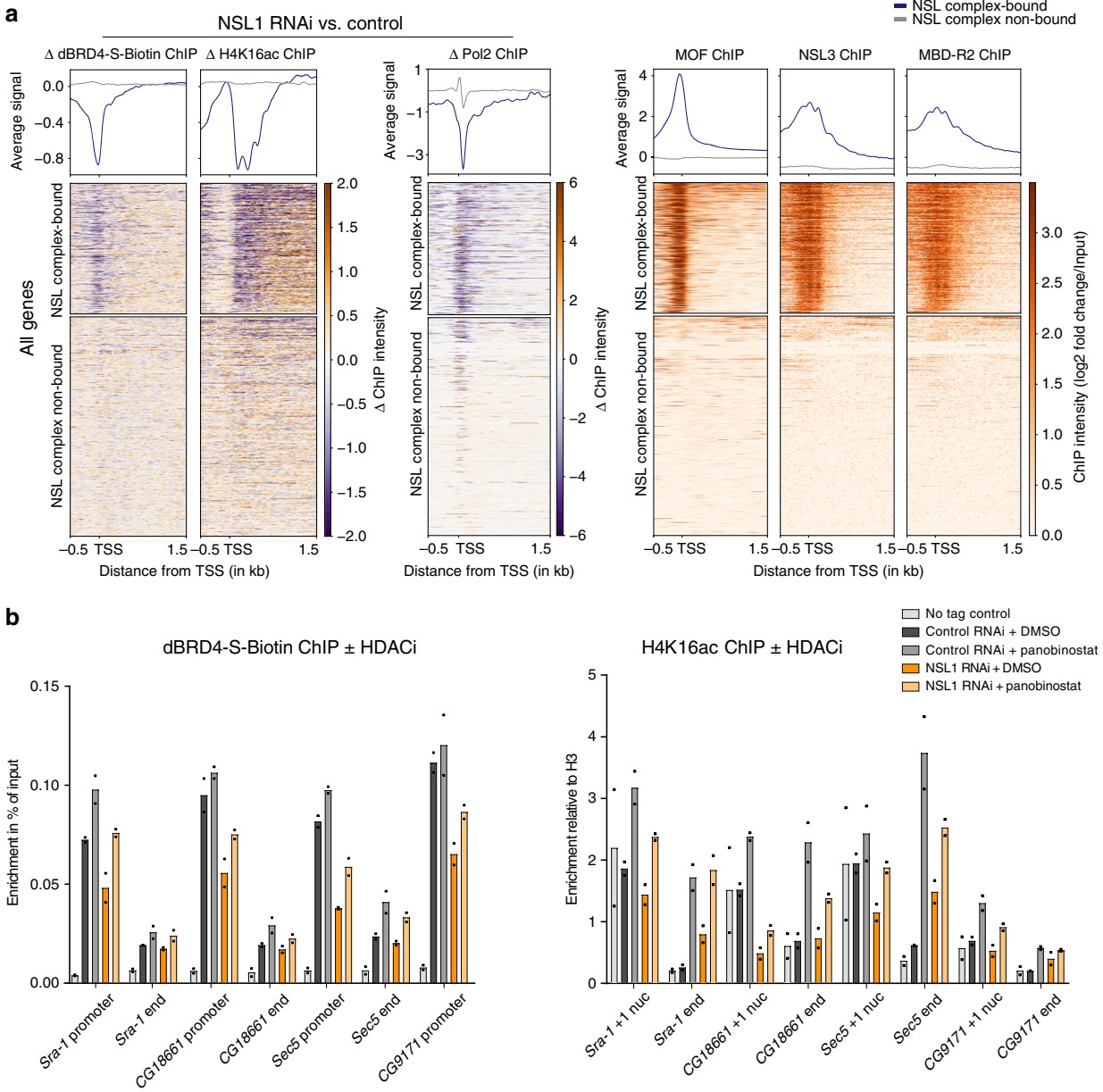

**Fig. 5 NSL complex-mediated acetylation leads to dBRD4 recruitment. a** Heatmaps and profile plots of ΔdBRD4-S-Biotin ChIP signal, ΔH4K16ac ChIP signal and ΔPol2 ChIP signal in control (GST) RNAi versus NSL1 RNAi[13], as well as ChIP-seq profiles of MOF, NSL3 and MBDR2[13]. All genes are plotted, clustered into two groups (clustering by k-means based on MOF, NSL3 and MBDR2 ChIP profiles). Order of genes is maintained for all other heatmaps in this panel. Signal of two merged replicates for ΔdBRD4-S-Biotin, ΔH4K16ac and ΔPol2 is shown. ΔdBRD4-S-Biotin was calculated as ChIP control-ChIP RNAi, ΔH4K16ac and ΔPol2 were calculated as log2 fold change (ChIP control over input) − log2 fold change (ChIP RNAi over input). **b** ChIP-qPCR of dBRD4-S-Biotin ChIP and H4K16ac ChIP after treatment with DMSO or HDAC inhibitor panobinostat (4 h, 200 nM) in control RNAi (GST) or NSL1 RNAi cells. Primers target the promoters and ends of dBRD4 and NSL complex-bound genes. Data are presented as mean value of $n = 2$ biological replicates. Source data are provided as a Source Data file.

H4K16ac at target promoters. The histone acetylation is in turn required for recruitment of BRD4.

Our genome-wide screen for functional NSL complex interaction partners identified a multifaceted functional interaction network. We find several factors and complexes known to be involved in transcription elongation: the PAF complex[26], Bre1 ubiquitin ligase[44] and the NELF complex[26]. In addition, acute depletion of dBRD4 lead to transcription elongation defects in *Drosophila*, similar to what has been reported in mammalian cells[33]. Some of the identified elongation factors, such as the PAF complex, have been shown to physically interact with BRD4[45].

We also identified the TIP60 complex, another chromatin-associated complex as putative functional interaction partner of the NSL complex. It is a large multi-subunit complex involved in chromatin remodelling, histone acetylation and incorporation of H2A.V, a histone variant which combines functions of mammalian H2A.Z and H2A.X[46]. Depletion of H2A.V alone markedly reduced NSL complex-mediated transcription and its incorporation at the 5′ end of genes is thought to facilitate transcription[47]. In addition to chromatin components, we identified proteins that have not been associated with a direct role in transcription, for example subunits of the SCF-slmb ubiquitin ligase, or the PP2A phosphatase complexes. Possibly their function for NSL-mediated

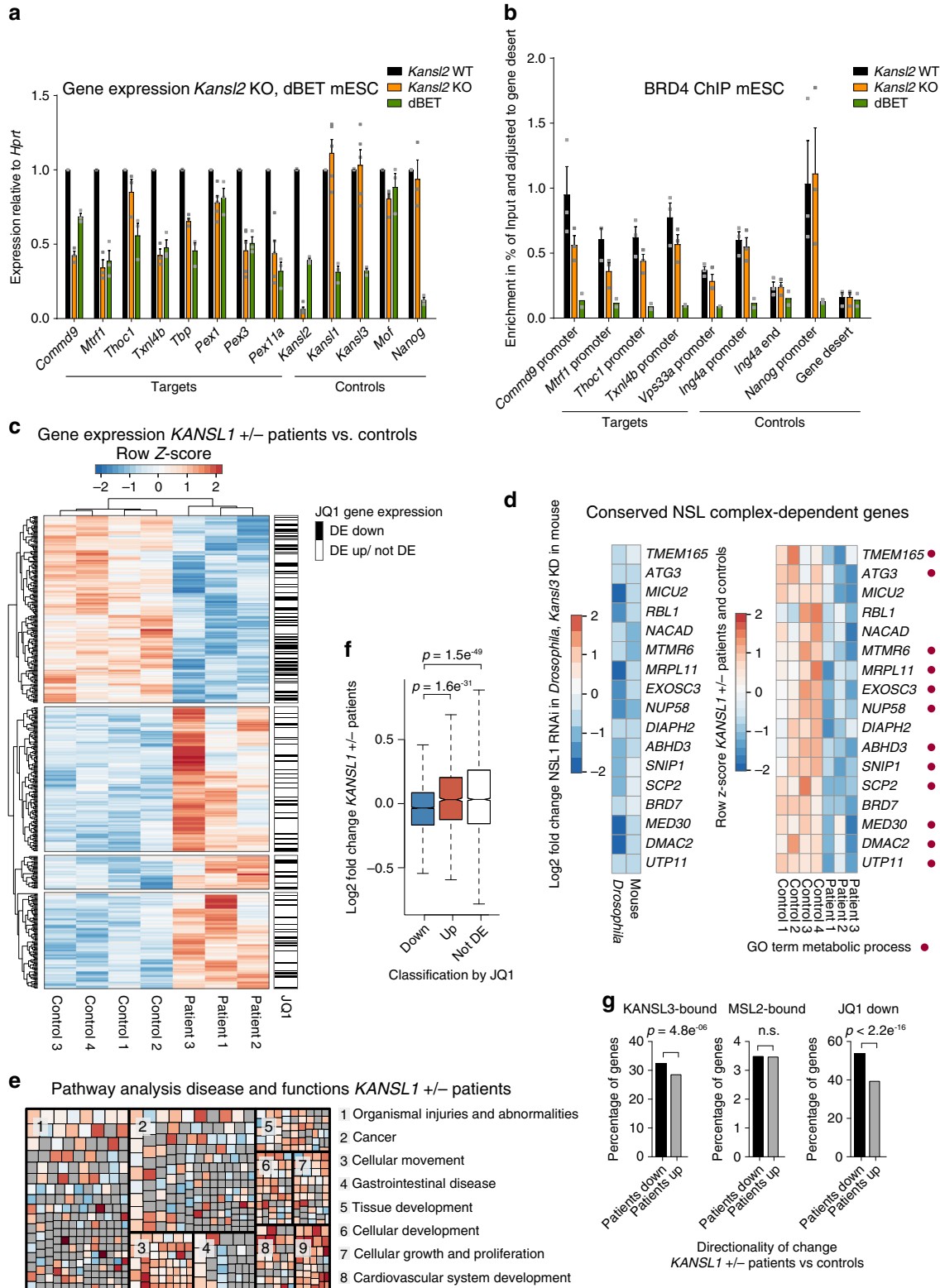

gene regulation lies rather in signalling to the nucleus or post-translational regulation of transcription factors. Our results suggest that the NSL complex acts as a chromatin-associated platform for integrating different signals within the cell. With the genome-wide screen we have generated a valuable resource. It will be interesting to dissect interconnections and hierarchies between the identified co-factors in future studies.

Evidence from crystal structures as well as biochemical data indicate that BRD4 targeting to chromatin occurs via its recognition of acetylated histone residues[36,37,48,49]. However, previous studies disagreed on which precise histone modifications are responsible for BRD4 recruitment[49–51]. Our data show that the NSL complex contributes to chromatin recruitment of BRD4 in *Drosophila* and mice through acetylation of the histone H4 tail.

**Fig. 6 The NSL complex/BRD4 axis is conserved in mice and humans. a** RT-qPCR analyses of *Kansl2*[fl/fl], Cre-ERT2 mESCs with (*Kansl2* KO) and without (*Kansl2* WT) tamoxifen treatment (500 nM, 3 days), and dBET6-treated (100 nM, 4 h) *Kansl2* WT cells, see also Supplementary Fig. 6. Expression is normalized to HPRT and relative to *Kansl2* WT. **b** ChIP-qPCR of BRD4 in *Kansl2* KO, *Kansl2* WT and dBET6-treated mESCs. ChIP enrichments are relative to gene desert signal. Primers target KANSL2-responsive (target) and KANSL2-non-responsive (control) genes. **a, b** Bars represent mean ± SEM (*n* = 3, for dBET ChIP *n* = 2 biological replicates). Source data are provided as a Source Data file. **c** Heatmap of row-scaled normalized RNA-seq counts from fibroblast cell lines of three Koolen-de Vries/*KANSL1* haploinsufficient (*KANSL1*+/−) patients and four controls (for genes FDR < 0.2). Order was generated by unsupervised hierarchical clustering. Genes significantly downregulated (DE down) (FDR < 0.05) upon JQ1 treatment in MOLT4 cells[33] are indicated in black, upregulated (DE up) or not differentially expressed (not DE) in white. **d** Left, log2 fold changes of conserved, NSL complex-dependent genes for NSL1 RNAi in *Drosophila* (FDR < 0.05), and *Kansl3* knockdown in mESCs (FDR < 0.05,[11]). Right, row-scaled normalized RNA-seq counts of *KANSL1*+/− patients and controls (FDR < 0.4). Red dot indicates association with GO term Metabolic Process. **e** Ingenuity Pathway Analysis showing most affected Disease and Functions groups of DE genes (FDR < 0.1) obtained from DESeq2 analysis of *KANSL1*+/− patient fibroblast RNA-seq. Grey Z-score indicates NA. Boxes are sized by negative log *p*-value. **f** Boxplots of log2 fold change of *KANSL1*+/− patient fibroblasts versus controls for classes of genes, based on differential expression upon JQ1 treatment[33]: DE down (*n* = 5581), DE up (*n* = 1482), not DE (*n* = 3716). Boxplots show median (centre), interquartile-range (box) and minima/maxima (whiskers).Two-sided Wilcoxon-rank-sum test was applied. **g** Left and middle: percentage of genes whose mouse orthologues are promoter-bound by KANSL3 (*n* = 3665)(left) or MSL2 (*n* = 846) (middle) in mESC and downregulated (black bars) or upregulated (grey bars) in *KANSL1*+/− patients. Right: percentage of genes significantly downregulated (FDR < 0.05) upon JQ1 treatment in MOLT4 cells (*n* = 5581)[33] and down-or upregulated in *KANSL1*+/− patients. Up/down classification of patient gene expression indicates directionality. Overrepresentation of downregulated genes was tested with one-sided Fisher's exact test.

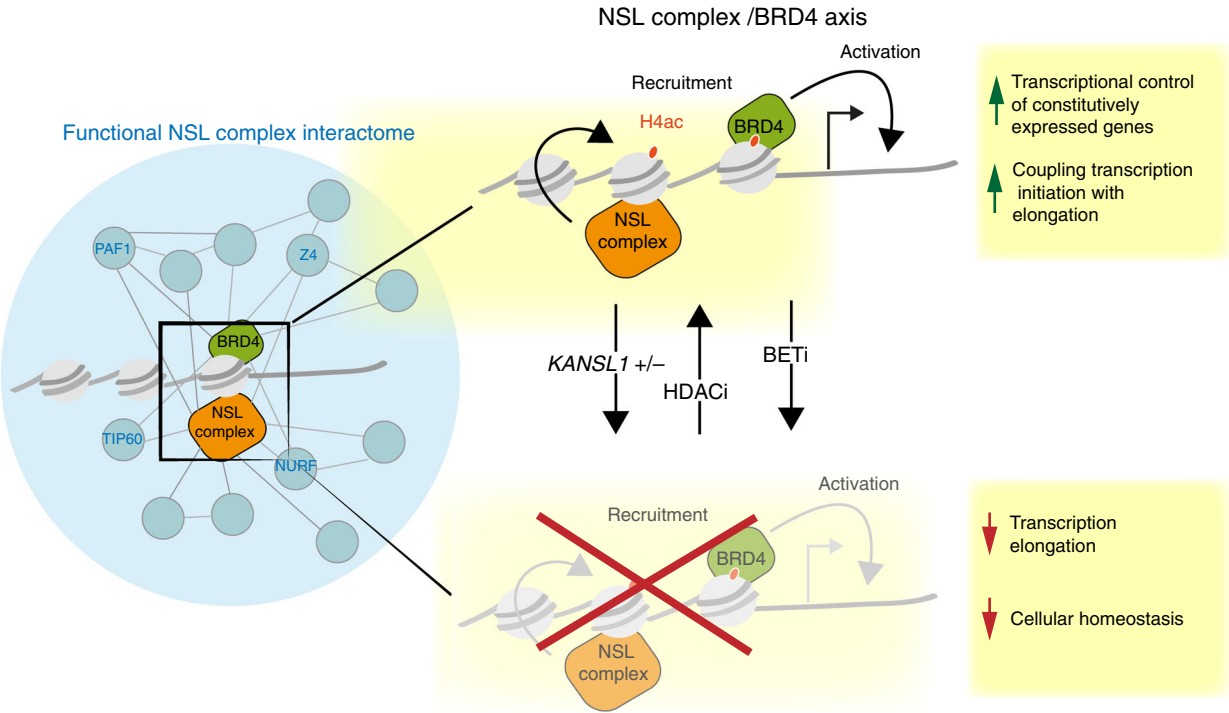

**Fig. 7 Model of transcriptional activation of NSL complex target genes.** Generating a functional NSL complex interactome (represented as a blue network, left) enabled identification of BRD4 as the key interaction partner of the NSL complex in driving transcription of essential genes in *Drosophila*, mice and humans. Molecular characterization of this interaction led us to propose the following model (right): The NSL complex recruits BRD4 to target promoters via acetylation of the histone H4 tail, while BRD4 is required for the transition of RNA Pol2 from transcription initiation to elongation. Disturbance of the NSL complex–BRD4 axis by BET inhibitors (BETi) or *KANSL1* haploinsufficiency (as seen in Koolen-de Vries syndrome) results in transcription elongation defects and compromised cellular homeostasis. BRD4 targeting defects can be restored by treatment with HDAC inhibitor (HDACi).

This is in agreement with a recent small-scale study in human cells showing that depletion of MOF and subsequent reduction in H4K16ac correlate with loss of BRD4 from promoters of several autophagy-related genes[50]. Nevertheless, since combinatorial effects of multiple acetylated histone residues enhance the affinity of BET proteins to histones (Supplementary Fig. 5 and refs. [36,38,48]), it is likely that additional acetyl-marks on the histone octamer, mediated by other histone acetyltransferases, such as P300, may also impact on chromatin recruitment of BRD4 and dBRD4.

Our work reveals the significant extent to which BRD4 contributes to expression of constitutive genes. Most previous analyses of BRD4 had focused almost exclusively on its role at developmental targets. The current hypothesis for the effect of BRD4 inhibitor treatment against leukemia is that the action of key developmental transcription factors, such as MYC, is blocked upon BRD4 inhibition[28,49,52,53]. Regulation of housekeeping genes by BRD4, however, raises the intriguing possibility that in addition to the known deregulation of key transcription factors, the sensitivity of leukemia cells to BRD4 inhibition could in part be explained by a generally higher demand of cancer cells for housekeeping gene products. Indeed, higher proliferation rates in cancer cells compared to healthy cells are connected to a higher need of metabolism and cell cycle-related proteins[54]. Thus,

inhibiting housekeeping gene transcription would render cancer cells more susceptible to treatment than healthy less proliferating cells. Consistently, a recent study identified MOF and BRD4 as important factors for *MLL-AF9* leukomogenesis[55]. It remains to be explored, if and how the misregulation of housekeeping genes affects cancer proliferation in the context of BRD4.

Our genome-wide analyses on patient fibroblasts and mouse embryonic stem cells indicate that reduction of the NSL complex at target gene promoters is associated with gene misregulation in *KANSL1*-haploinsufficient patients. Although this molecular causality is implied[56], there is currently no targeted medical treatment available for the Koolen-de Vries syndrome. Given that expression data from patients revealed a BET protein signature, and dBRD4 targeting defects could be rescued by HDACi treatments, it is tempting to speculate that restoring acetylation levels in patients could possibly revert gene misexpression and consequently alleviate some of the symptoms associated with the Koolen-de Vries syndrome.

HDACi treatment, when tested in the context of diseases linked to acetylation imbalance, has already given promising outcome in cellular model systems[57] and was effective against epilepsy in patients carrying KAT8 mutations[9]. In future, it would be important to determine the contribution of BRD4 to the cellular response to HDACi.

Our study uncovers a conserved crosstalk between the NSL complex and BRD4 for the transcription regulation of a diverse repertoire of common target genes. This crosstalk links a housekeeping gene regulator (NSL complex) to a well-known, disease-relevant protein (BRD4). This, together with our expression analysis from Koolen-de Vries patients, puts forward the concept that the breadth of constitutive gene functions confers disease relevance.

## Methods

**Cell culture**. *Drosophila* S2 cells (Thermo Fisher R69007) were cultured at 25 °C in serum free medium (Express Five SFM, Gibco), supplemented with 20 mM L-Glutamin (GlutaMAX, Gibco). Cells were passaged with 10% of conditioned medium each passage. Cells were maintained at a density of 1–12 million ml$^{-1}$.

Mouse embryonic stem cells were cultured in a humidified incubator at 37 °C and 5% CO$_2$. Cells were grown on Attachment Factor (Thermo Fisher) in 2i medium (KnockOut$^{TM}$ Dulbecco's modified Eagle's medium (DMEM) (Gibco) supplemented with 15% KnockOut$^{TM}$ Serum Replacement (Gibco), 0.1 mM MEAA (Gibco, non-essential aminoacid), 1 nM Na-Pyruvate (Gibco), 0.1 mM 2-mercaptoethanol (Gibco), 4 mM L-Glutamin (GlutaMAX, Gibco), 5 µg ml$^{-1}$ Insulin (Sigma), 50 U ml$^{-1}$ Pen-Strep, 200 U ml$^{-1}$ LIF (ESGRO), 1 µM PD0325901 (StemGent), 3 µM CHIR99021 (StemGent).

Human fibroblast cell lines were established from skin biopsies of confirmed *Koolen de-Vries* syndrome patients and healthy controls. Cells were maintained in in a humidified incubator at 37 °C and 5% CO2 in DMEM (GlutaMAX supplement, Life Technologies), 100 U ml$^{-1}$ penicillin, 100 µg ml$^{-1}$ streptomycin and 10% foetal calf serum (FCS). All fibroblast lines tested negative for Hepatitis B, Hepatitis C, HIV and human Herpes virus 4 and 8. Fibroblasts were passaged at ~90% confluency.

**Human research participants**. The affected individual and/or their families gave written consent for their inclusion in the analysis under two local ethic committees —approved protocol (S60206) to Leuven University for control samples and (DC-2008-735) to Université Grenoble-Alpes CHU for patient specimens.

Distribution of human samples used in the study is restricted, as consent from the patient families is required for further use.

**Cloning**. Plasmids were generated by PCR-based restriction cloning. PCR amplification was performed using PrimeSTAR GXL DNA Polymerase (Takara). For dBRD4-S-stable cell line, cDNA encoding for dBRD4-S was PCR amplified from the Drosophila Gold Collection clone LD26482 (available though DGRC), using primers with PacI and AscI restriction site overhangs (see Supplementary Data 3) and cloned into a modified pCo-Blast vector (Thermo Fisher K5150-01), containing a copper inducible MtnA promoter and a 3xFLAG-6xHis-Bio-6xHis cassette for C-terminal tagging. For recombinant dBRD4-S expression, dBRD4-S (aminoacids 1-1063) was amplified from cDNA as for dBRD4-S stable cell line (see above), using primers with BamHI and HindIII restriction sites (see Supplementary Data 3) and cloned into a customized pET28 vector containing a 6xHis-SMT3

cassette for N-terminal tagging. The vector for expression of the *Schizosaccharomyces pombe* GST-HP1$^{SW16}$ was constructed in a pGEX backbone.

**Generation of dBRD4-S stable cell line**. The dBRD4-S-Biotin pCo-Blast based plasmid (see cloning) was transfected in S2 cells using effectene transfection reagent (Qiagen), transfected cells were selected with 5 µg ml$^{-1}$ Blasticidin (Gibco) for 10 days. Expression of dBRD4-S-Biotin was induced by addition of CuSO$_4$ (1 mM final) for 16 h.

**Mouse ethics statement**. Animals were kept on a 12:12 h light–dark cycle and provided with food and water ad libitum. Animal husbandry and all experiments were approved by the committee on ethics of animal experiments of the state Baden-Württemberg (Regierungspräsidium Freiburg).

**Generation of mouse embryonic stem cells**. Mouse embryonic stem cells were derived from 8-weeks-old natural-timed, mated, pregnant females by immuno-surgery according to[58]. Briefly, *Kansl2*$^{fl/fl}$ (tm1a(EUCOMM)Wtsi, purchased from the international knockout mouse consortium (IKMC)) females were crossed with *Kansl2*$^{+/fl}$, CAG::CreERT2 (purchased from Jackson laboratory) males. Both strains had been backcrossed with C57Bl/6J. E 3.0 dpc embryos were flushed from the uterus and placed into drops of pre-equilibrated EmbryoMax KSOM medium (Merck) supplemented with 0.1 mM MEAA (Gibco, non-essential aminoacid), 1 µM PD0325901 (StemGent) and 3 µM CHIR99021 (StemGent). Acid tyrode was used to remove the zona pellucida. Embryos were then transferred to wells of a 4-well plate containing KnockOut$^{TM}$ DMEM (Gibco) supplemented with 1 µM PD0325901 (StemGent), 3 µM CHIR99021 (StemGent), 200 U ml$^{-1}$ LIF (ESGRO) and 20% rabbit anti-mouse serum (Sigma # M5774) and incubated for 1 h in a humidified incubator at 37 °C and 5% CO2. Excess serum was removed by media washes (KnockOut$^{TM}$ DMEM (Gibco) supplemented with 1 µM PD0325901 (StemGent), 3 µM CHIR99021 (StemGent), 200 U ml$^{-1}$ LIF (ESGRO) and the embryos were transferred for 30 minutes into droplets containing KnockOut$^{TM}$ DMEM (Gibco) supplemented with 1 µM PD0325901 (StemGent), 3 µM CHIR99021 (StemGent) and 20% of freshly thawed complement sera from guinea pig (Merck, #234395). Following careful washes, trophectoderm was separated from the epiblast by pipetting up and down with a fine Pasteur pipette (smaller than the inner cell mass (ICM)). Isolated epiblasts were cultured on gelatin-coated plates in 2i medium (see Cell Culture subsection above). When the epiblasts had reached their optimal size, they were disaggregated using Accutase (Sigma) and transferred using a fine Pasteur pipette evaluating separation efficiency at a stereomicroscope.

***Drosophila* husbandry**. *Drosophila melanogaster* were reared on a standard cornmeal fly medium at 25 °C, 70% relative humidity and 12 h dark/12 h light cycle. Experimental crosses for dBRD4-L-HA fly lines were conducted at RT. The following stocks were used:

w$^{1118}$;P{da-GAL4.w-}3 (Bloomington stock #8641)
w;; UAS-Fs(1)h.L::HA (Bohmann laboratory)

**RNAi in *Drosophila* S2 cells**. Production of dsRNAs was performed in the following way: DNA template for in vitro transcription was generated by PCR (primers as listed in Supplementary Data 3), T7 in vitro transcription was carried out using HiScribe T7 High Yield RNA Synthesis Kit (NEB) according to manufacturers' instruction. RNA cleanup was performed using MEGAclear Transcription Clean-Up Kit (Ambion). in all, 10 µg dsRNA were used per 1 × 10$^6$ cells. Cells were harvested after 4 days. This method was adapted from a previously described method[59].

**Drug treatments**. Drug treatments were performed by adding the respective reagent to the cell culture medium; timings as indicated in the figure legends. The following inhibitors were used: JQ1 (1187, Cayman Chemicals, 5 µM), iBET 762 (10676, Cayman Chemicals, 1 µM), dBET6(Gray lab, 100 nM), Panobinostat LBH-589 (404950-80-7, Biozol Diagnostica, 200 nM).

**RNAi screen**. HD2.0 library of dsRNAs was used. This library was designed using the software NEXT-RNAi[18], excluding low complexity regions and sequence motifs inducing off-target effects. 14000 protein coding genes and 1000 non-coding genes are targeted. The genome-wide RNAi screen was performed with two technical replicates. Per well 1.4 × 10$^4$ cells were seeded in 384-well plates prespotted with 250 ng dsRNAs each. After 24 h cells were transfected with a plasmid mix of 3 ng Renilla reporter (pRL-hsp70), 17.5 ng UAS-firefly reporter (pG5luc) and 0.87 ng NSL3-Gal4 DBD activator (pAct5.1-NSL3-Gal4-DBD)[15] per well using effectene transfection reagent (Qiagen) according to manufacturer's instructions. 3d post transfection, cells were lysed and subjected to dual luciferase assay readout by chemiluminescence with a Mithras LB940 plate reader (Berthold Technologies). Each assay plate contained wells with positive control dsRNAs (NSL complex members: NSL1, NSL3 and MOF), and negative control dsRNAs (GFP, GST, Diap-1). Data were analysed using the online software CellHTS2[60] (http://web-cellhts2.dkfz.de/cellHTS-java/cellHTS2/) version 2.16.0 with the following settings:

normalization method: plate median; values not log transformed; summarize values: by mean; normalization scaling method: multiplicative; variance adjustment: no scaling; no viability function included. RNAi candidates with a change in Renilla of more than two fold up or down (Supplementary Fig. 1g) were excluded from further analysis and also from the subset library tested in the secondary screen.

Secondary screen was performed like genome-wide screen, but with a subset library consisting of 367 candidates, 302 with Z-score < −7.32 (NSL co-activators) 37 with Z-score > 3 (NSL antagonists) and 28 additional candidates of interest.

**Luciferase assays (96-well plate format).** Luciferase assays were performed as in the genome-wide RNAi screen, but seeding $1 \times 10^5$ cells per well in a 96-well plate prespotted with 1 µg dsRNA and transfecting a plasmid mix of 17 ng Renilla reporter (pRL-hsp70), 100 ng UAS-Firefly reporter (pG5luc) and 3.12 ng nsl3-Gal4-DBD activator (pAct5.1-nsl3-Gal4-DBD) per well, three technical replicates per assay. For luciferase assays with inhibitor treatments, at time of treatment 10×(of final) concentrated inhibitor or DMSO containing medium was added to the cell culture medium. Readout was performed by chemi luminescence using the Dual Luciferase Assay Kit (Promega) and a Centro LB plate reader (Berthold Technologies).

**Biolayer interferometry measurements.** GST-HP1$^{SWI6}$ and SMT3-dBRD4-S proteins were recombinantly produced in Rosetta$^{TM}$ (DE3) Competent Cells by induction with 0.25 mM IPTG at 20 °C overnight for HP1$^{SWI6}$ and 1.5 h at 37 °C for dBRD4-S. Bacteria were collected by centrifugation and flash frozen. For biolayer interferometry measurements bacteria were lysed by sonication in lysis buffer (100 mM NaCl, 25 mM HEPES pH7.4, 2 mM beta-Mercaptoethanol, 0.1% Tween20, 1x Complete Protease Inhibitors (Roche)) followed by centrifugation at 20000 rpm for 30 min. The soluble lysate was directly used for Bio-Layer Interferometry (BLI) measurements using a BLItz instrument (ForteBio) with the Advanced Kinetics program and Dip and Read$^{TM}$ High Precision Streptavidin (SAX) Biosensors. Total protein amounts per lysate (uninduced, GST-HP1$^{SWI6}$ and SMT3-dBRD4-S) were measured by Bradford and adjusted. N-terminally biotinylated histone H4 peptides (aminoacids 1-23) harboring acetylated lysine residues were purchased from EpiCypher (H4 unmodified #12-0029, H4K16ac #12-0033, H4K12ac # 12-0032, H4K12,K16ac # 12-0136, H4K5,8,12,K16ac # 12-0034) and used at 1 µM concentration. The measurements were carried out using lysates from uninduced bacteria as baseline. The signal from H4 unmodified peptide measurements was subtracted from all experimental measurements. As control, the H4 ac peptide measurements were also performed with HP1$^{SWI6}$ lysates, which were previously tested to display binding to H3K9me3 modified peptides.

**Immunoblotting.** Proteins were separated by SDS-PAGE, transferred at 110 V to a PVDF membrane in Tris-Glycine Transfer-Buffer. Membranes were blocked for 30min-1h in 5% BSA in TBS-0.1% Tween, then incubated with primary antibodies overnight at 4 °C. Secondary HRP-coupled antibodies were used at 1:10 000 dilution for 1 h. Blots were developed using Lumi-Light Western Blotting substrate (Roche) and imaged on a ChemiDoc XRS + machine (Biorad). Following primary antibodies were used: dBRD4 (Paro lab, ID166, 1:1000), Pol2 ser2p (ab5095, Abcam, 1:5000), Pol2 ser5p (ab5131, Abcam, 1:5000), Rpb3 (Akhtar lab, 1:1000), H3 (ab10799, Abcam, 1:3000), NSL3 (Akhtar lab, 1:1000), MOF (Akhtar lab, 1:3000), H2A.V (61752, Active Motif, 1:2000), MCRS2 (Akhtar lab, 1:3000), H4K16ac (07-329, Millipore, 1:3000), KANSL2 (HPA038497, Sigma, 1:1000), BRD4(A301-985, Bethyl, 1:2000), ACTIN-HRP(sc-1616, Santa Cruz, 1:5000), GAPDH-HRP (MA5-15738, Thermo Scientific, 1:5000), FLAG-HRP(A8592, Sigma, 1:5000). Uncropped immunoblots are provided in Supplementary Figure 8a–g.

**RNA expression analysis.** For transcriptomic analyses of Koolen de-Vries patient and control fibroblasts, cells were grown in parallel with each biological replicate representing a fibroblast cell line derived from a different individual. Three independent fibroblast cell lines from Koolen de-Vries patients and four from healthy individuals were used between passages 6 and 8. RNA was isolated using the RNeasy kit (Qiagen) and libraries prepared using the Illumina TruSeq preparation kit.

For Drosophila RNA seq experiments three biological replicates were conducted with one passage difference between each of the replicates. RNA was extracted using a Trizol based kit (Directzol, Zymo), according to the manufacturer's manual. Sequencing libraries were prepared according to Illumina's TruSeq Stranded Total RNA Library Prep protocol for total RNA sequencing. Relative to total RNA amounts per sample, ERCC spikes were added before library preparation. For NSL1 depletion, the ratio of total RNA/ total DNA per sample did not change over the course of the experiment. Since the DNA amount per cell should not change during a knockdown experiment, we use ERCC spike normalized data as approximation of true transcript abundance for the Drosophila RNA-seq data in this study. Libraries were sequenced on a HiSeq2500 (Illumina) with a sequencing depth of $30 \times 10^6$ reads per sample (Drosophila samples) and $40 \times 10^6$ reads per sample (fibroblast samples).

For RT-qPCRs, cDNA was synthesized using the GoScript$^{TM}$ Reverse Transcription System (Promega) with random primers for all Drosophila experiments and oligo(dT) primers for experiments from mammalian cells from 0.5–1 µg of total RNA, for mESC cDNA 10% Drosophila total RNA were added prior to RT reaction. Expression of target genes was normalized to robustly not changing genes in RT-qPCR experiments (HPRT for mESC, ATPsynbeta, sun and SP1029 for Drosophila cells)

**Quantitative real-time PCR.** qPCR was performed using FastStart Universal SYBR Green Mastermix (Roche) on a Roche LightCycler 480 with 300 nM final primer concentration in 7 µl reaction volume. We corrected for primer efficiency using serial dilutions.

**RNA-seq analysis.** For Drosophila samples, reads were mapped to the Drosophila genome (dm6) using subreads[61], transcripts were counted with htseq[62] and differential expression analysis was performed using DESeq2[63]. Here ERCC spike normalization was performed. ERCC spikes were added according to total RNA amount per sample before library preparation. Spikes were mapped using STAR[64] (default parameters) and counted using feature counts[65] (default parameters). DESeq2 on spike counts (again of three biological replicates per sample) was used to calculate the size factor, which was used to normalize gene expression DESeq2. Expression analysis by qPCR of another independent replicate was performed to validate the RNA-seq results. RNA seq from fibroblast samples was processed using Snakepipes. Reads were mapped to hg38 genome using STAR (default parameters) and transcripts were counted using featureCounts. Differential expression analysis was performed with DESeq2[63], comparing healthy control samples ($n = 4$) to patient samples ($n = 3$). Pathway analysis for patient fibroblast samples were performed using Ingenuity Pathway Analysis (Qiagen).

**Chromatin Immunoprecipitation (ChIP) and library preparation.** Drosophila S2 cells were fixed for 10 min in 1.8% of formaldehyde at 23 °C shaking. Fixation was quenched by the addition of glycine (0.125 M). Nuclei were enriched by the following wash steps (all washes 5 min each at 4 °C) 2x buffer A (10 mM HEPES pH 7.6, 10 mM EDTA, 0.5 mM EGTA, 0.25% Triton X-100, protease inhibitors), 3x buffer B (200 mM NaCl, 10 mM HEPES pH 7.6, 1 mM EDTA, 0.5 mM EGTA, 0.01% Triton-X 100, protease inhibitors) and 3x RIPA buffer (25 mM HEPES pH 7.6, 150 mM NaCl, 1 mM EDTA, 1% Triton-X 100, 0.1% SDS, 0.1% DOC, protease inhibitors). Sonication was performed in RIPA buffer, first 9 × 20 s using a Branson 250 sonicator at 40 pulse, intensity 2.5, then 12 min using a Covaris soncator E220 (following settings: peak power 150, duty factor 10, cycles/burst 200), yielding fragment sizes of 250–500pb. After clearing the chromatin extract (10 min, 12000 g), supernatant was used for IP. Biotin ChIP experiments were performed as described by[66], with slight modifications. In brief, to 200 µg chromatin per IP were diluted in RIPA buffer and added to 20 µl of streptavidin T1 magnetic beads (Invitrogen) previously blocked for 1 h with 1% cold fish skin gelatin (Sigma Aldrich) and 1 mg yeast tRNA (Ambion). Per 200 µg chromatin (as measured from DNA absorption) × 130 pg biotin labeled DNA fragments were added to the IP. Three different biotin spikes were generated by reverse transcription of ERCC RNA spikes and amplification with biotin labeled primers. Chromatin was incubated with streptavidin magnetic beads for at least 6 h at 4 °C. Beads were washed for 8 min at RT for each wash step, two washes in SDS wash buffer (2% SDS in TE buffer (10 mM Tris-HCl, 1 mM EDTA)), 1 wash in high salt RIPA buffer (RIPA buffer with 500 mM NaCl instead of 140 mM), 1 wash in LiCl buffer (250 mM LiCl, 10 mM Tris-HCl, 1 mM EDTA, 0.5% NP-40, 0.5% DOC), two washes in TE buffer. Beads were reverse crosslinked overnight at 65 °C followed by RNAseA (Thermo Scientific, 0.1 mg ml⁻¹) (30 min 37 °C) and proteinase K digestion (Ambion, 0.1 mg ml⁻¹) for (2 h at 55 °C). DNA was purified using minelute columns (Qiagen). For ChIP of endogenous proteins 4 µl of NSL1, MOF, dBRD4 or Rbp3 antibody were added to 10 µg chromatin, diluted in RIPA buffer and incubated at 4 °C overnight. For Rbp3 ChIPs additional 2 µg of sheared Drosophila virilis chromatin were added to each IP to be able to control for IP efficiencies. 20 µl blocked (as streptavidin beads) magnetic protein A/G beads (Invitrogen) were used to capture immunocomplexes (2 h at 4 °C). Washes of beads were done at 4 °C for 3 × 10 min in RIPA buffer, 1 × 10 min in LiCl buffer, 1x TE buffer. Reverse crosslinking, RNAse and ProtK digestion were done like for biotin ChIP, see above.

For ChIP in mESCs, cells were fixed for 10 min in 1.8% formaldehyde at 23 °C shaking and quenched by addition of glycine (0.125 M). Isolation of nuclei was performed like as for Drosophila cells. Then nuclei were sheared for 20 min in 10 mM Tris (pH 8.0), 100 mM NaCl, 1 mM EDTA, 0.5 mM EGTA, 0.1% DOC, 0.5% sodium dodecyl sarcosinate, 1% Triton X-100 [29] using a Covaris soncator E220 (following settings: peak power 105, duty factor 2, cycles/burst 200). IP, washes, reverse crosslinking and cleanup of DNA was performed as for Drosophila endogenous antibody ChIPs. ChIP-QPCR recovery was determined as the amount of immunoprecipitated DNA relative to input DNA.

ChIP-sequencing libraries were prepared from 0.5–1 ng (for biotin IP) 2–10 ng (for histone and Rpb3 IPs) using NEBNext Ultra2 Library Preparation Kit, according to manufacturers' manual, paired end sequencing was done on a HiSeq3000 (Illumina), 75 bp read length, sequencing depth of ca. 10mio reads per sample.

**ChIP-seq data analysis**. Sequencing reads of ChIP samples and their respective input samples were trimmed using TrimGalore (developed by Felix Krueger, Babraham Institute) (quality threshold of 20), subsequently the reads were mapped to the *Drosophila* genome (dm6) using Bowtie2[67] (default settings, no duplicate removal). After mapping the BAM files of the two biological replicates of each experiment were merged. If not indicated differently coverage of ChIP data was normalized to input data by calculating log2 fold change of ChIP/input using bamCompare and bamCoverage from deepTools2[68]. For visualization of ChIP seq tracks IGV was used. For Rpb3 ChIPs the ChIP/input enrichment was calculated using Signal Extraction Scaling (SES) method. Enrichment of ChIP signal on promoters ($\pm$200bp from TSS for transcription factors +500 bp for histone marks) was generated using MultiBigwigSummary tool from deepTools2[68].

Biotin ChIP efficiencies were assessed by mapping and counting the number of reads mapping to a spike annotation for each sample (Bowtie2). Similarly Rpb3 ChIP efficiencies were assessed by mapping and counting the number of reads assigned to the *Drosophila virilis* genome (Bowtie2, dvir1.3). As no major differences in external spike numbers (biotin or *Drosophila virilis* spikes) could be observed between the samples, standard normalization methods (log2 fold change ChIP/input) were applied.

To generate the list of dBRD4-S-Biotin-bound genes, peaks of endogenous dBRD4 ChIP and dBRD4-S-Biotin ChIP were called using MACS2[69] and the following parameters: band width, 300; model fold, 5–50; $p$-value cut-off, $5 \times 10^{-5}$; both datasets were generated in this study. Whenever the called peak region overlapped with TSS $\pm$ 200 bp the gene was defined as bound gene. The dBRD4-S-Biotin-bound genes were only considered if the gene was also bound by endogenous dBRD4.

Similarly, to calculate percentages of dBRD4 and NSL3 co-bound promoters, peaks of endogenous dBRD4 ChIP[25] and NSL3 ChIP[13] were called using MACS2 (model fold, 10–30; $p$-value cut-off, $1 \times 10^{-5}$ and model fold, 5 to 50; $p$-value cut-off, $5 \times 10^{-5}$). Whenever the called peak region overlapped with TSS $\pm$ 200 bp the gene was defined as bound gene. The bound gene lists were then compared to calculate overlaps. Gene Ontology (GO) Enrichment Analysis was conducted with the GO Ontology database (Released 2019-07-01) using PANTHER Overrepresentation test (Released 2019-07-11) with Fisher analysis on http://geneontology.org/ and visualized using REVIGO (http://revigo.irb.hr/r).

**Generating NSL complex-bound and non-bound gene lists**. To generate the lists of expressed, NSL complex-bound and non bound genes (Fig. 3) published NSL3 and MBDR2 profiles[13] were analysed similar as above. Genes were filtered for expressed genes (more than 10counts of mean expression in control RNAi RNAseq dataset (this study)). Enrichment of ChIP signal of the promoter regions ($\pm$500 bp from TSS) was generated using MultiBigwigSummary (deepTools2)[68]. Genes were ranked according to NSL3 binding, based on visual inspection of grouped binding profiles, a cutoff to distinguish bound from non-bound expressed genes was chosen, such that lowly-bound genes were excluded from the non-bound gene list. As the primary aim was to generate a stringent list of non NSL-bound expressed genes, rather than a stringent list of NSL-bound genes. MBDR2 binding profiles were used to verify that the lists represent NSL complex-bound and non-bound genes.

**Polytene chromosome immunostainings**. Polytene stainings of salivary glands of wandering third instar larvae were carried out as follows. Salivary glands were dissected, fixed for 7 min in fixation solution (1.8% Formaldehyde, 0.45% acetic acid), then transferred to a polylysine coated slide and covered with a coverslip. Spreading of chromosomes was achieved by applying the tip of a 200 rpm Dremel and subsequent flattening was achieved with the use of a MTC-300-1 vice (Avenger Gold Toolmaker). Slides were blocked with 10% goat serum (Invitrogen), followed by primary and secondary antibody incubations. Primary antibodies were used at the following concentrations: anti HA (Covance) 1/400, anti NSL3 1/250, Secondary Antibodies were used at 1/500 dilution. Images were captured on a LSM780 confocal microscope (Carl Zeiss Microscopy) using an alpha Plan-Apochromat 63×/1.4 (DIC) Oil objective. This method was adapted from a previously described method[70].

**Generating list of conserved NSL complex-dependent genes**. Genes downregulated following NSL1 RNAi in *Drosophila* cells (FDR < 0.05), *Kansl3* shRNA treatment in mESC[11] (FDR < 0.05) and genes downregulated in Koolen-de Vries patients (FDR < 0.4) were overlapped. Orthologue predictions were obtained from Ensembl BioMart.

**Data from public repositories**. From NCBI gene expression omnibus (GEO) (http://www.ncbi.nlm.nih.gov/geo/)
dBRD4 ChIP-seq from S2 cells: GSE36450
MOF ChIP-seq: GSE37864
KANSL3, MCRS1, MOF ChIP-seq in mESC: GSE51746
*Kansl3* knockdown RNA-seq in mESC: GSE57698
BRD4 ChIP-seq: mESC and EB: GSE76760
BRD4 ChIP-seq: MLL-AF9: GSE74536
JQ1 RNA-seq MOLT4 cells: GSE79253

From ArrayExpress
NSL3 and MBD-R2 ChIP-seq from S2 cells: E-MTAB-1085
Pol II ChIP-seq from S2 cells (GFP RNAi, NSL1 RNAi): E-MTAB-1084

**Reporting summary**. Further information on experimental design and reagents can be found in the Nature Research Reporting Summary linked to this article.

## Data availability
RNA-seq and ChIP-seq data have been deposited to Gene Expression Omnibus (GEO) under the accession number: GSE135815 and are also accessible under BioProject PRJNA560185. All other relevant data supporting the key findings of this study are available within the article and its Supplementary Information files or from the corresponding author upon reasonable request.

The source data underlying Figs. 2a, 3b–d, 4d–g, 5b, 6a,b and Supplementary Figs. 1a, b, 3a–c, 4b, 4g,h, 5f,g, 6c–e, 7a,b are provided as Source Data files.

## Code availability
All data analyses used were performed using published software. Requests can be sent to the corresponding author.

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

## Acknowledgements
We thank Claudia Blass and Barbara Schmitt for technical assistance and Ulrike Hardeland for advice in performing the high-throughput screening. We thank Renato Paro for kindly sharing dBRD4 antibodies with us, Nathanael Gray and Stirling Churchman for providing support concerning the dBET compound and Dirk Bohmann for sharing the UAS-dBRD4-L-HA fly line. We thank members of the MPI-IE facilities for support. We thank Claudia Keller Valsecchi for help with biolayer interferometry experiments. We thank Plamen Georgiev and Anastasios Alexiadis for help with fly work and Nhuong Nguyen for generating mammalian luciferase constructs. We thank Devon Ryan, Fidel Ramirez, Friederike Duendar, Olga Bondareva and Gina Renschler for their support in bioinformatics analyses and Sukanya Guhathakurta for help with mESC generation and fibroblast cell culture. We thank Claudia Keller Valsecchi and Ken Lam for their valuable advice to the project. We thank Claudia Keller Valsecchi and Maria Shvedunova for critical reading and editing of the manuscript. We apologize to the authors of articles we did not cite due to space limitations. This work was supported by DFG funded CRC992, CRC1140 awarded to A.A. and DFG (DRIC infrastructure grant) awarded to MB. This study was supported by the German Research Foundation under Germany's Excellence Strategy (CIBSS—EXC-2189—Project ID 390939984).

## Author contributions
A.G. and A.A. conceptualized this study. M.B. supervised the functional screen. B.S. performed fibroblast RNA-seq experiments. M.F.B. generated mESC lines. M.V., M.N., C.C. and J.T. reviewed patient phenotypes. M.V., M.N., C.C., M.J.B. and J.T. performed sample collection and generated human fibroblast cell cultures. J.B. provided dBET reagents. A.G. planned and performed all other experiments, computational analyses and wrote the paper with input from A.A., M.F.B., B.S., M.B. and J.T. All authors reviewed, edited and approved the paper. A.A. supervised the study and provided guidance.

## Competing interests
The authors declare no competing interests.
