## [Peer Review File · Nature Communications]

Reviewers' comments:

Reviewer #1 (Remarks to the Author):

The manuscript by Gaub et al. describes a wealth of technically advanced experiments documenting a functional relationship between the histone acetyltransferase complex NSL and the acetyl-binding protein Brd4. The topic of the paper is of considerable interest since it deals with a novel pathway through which site-specific acetylation affect transcription.

Data obtained largely from the *Drosophila* model suggest a hierarchical scenario, according to which the H4K16 acetylation placed by NSL is bound by Brd4. Polymerase occupancy data suggest that early steps of transcription elongation may be promoted by Brd4, in keeping with the literature. Since all mentioned proteins are conserved in mammals, the authors apply the insights derived from the *Drosophila* model to human patients, in which the human NSL complex is impaired.

In some places the authors overstate their case. It seems that in *Drosophila*, NSL-mediated acetylation is one of several pathways to recruit Brd4. For the human data the situation is more complicated. I am not convinced that the manuscript keeps what is promised in the title. Below, I list observations mainly in chronological order of perceived shortcomings that should be addressed or clarified in a revised manuscript.

1. Fig.1: The screen is done well. Fig.1f may list Brd4 along with the complexes, for reference.
2. Fig.2c: These polytene chromosome stainings are not informative due to lack of resolution and can be omitted.
3. Fig.2b does not really illustrate the statement in the text, namely that 86% of NSL3-bound promoters are occupied by Brd4. Perhaps a scatter-plot is better suited, where correlations of signal strengths for individual sites are put in relation?
4. Fig.Suppl.3: Luciferase signals are appropriately normalized to a 'renilla' control. This works if the renilla signal does not fluctuate massively, or systematically. In Fig.S3a-c it is of concern that the renilla signal systematically increases in the presence of Brd4 inhibitors, so normalization distorts the picture. The heat shock promoter apparently was a bad choice. Is there an explanation? How many promoters are positively affect by BET inhibitors?
5. Fig.3f: The gene expression of NSL complex-bound genes are shown. What is missing is a control of an equivalent number of NSL-unbound genes, to establish the specificity.
6. Fig.3g: The correlation of r^2 of 0.22 is not a strong correlation as stated in the text, but at best moderate (albeit significant).
7. Fig.Suppl.3h: This is a very important control experiment, which is why the lower panel needs to show that the Brd4 RNAi has indeed worked.
8. Fig.3h,i: These effects should be quantified.
9. Fig.Suppl.4a lacks documentation that the NSL1 RNAi worked.
10. Fig.5a: The K16ac ChIP visually appears as if there was an enhanced signal in gene bodies upon NSL1 RNAi, in NSL-bound genes. Is this the case and how could it be explained?
11. Fig.5,,Suppl.5: The effect of HDACi are mainly interpreted in the context of H4K16ac levels, but 5b (NSL1 RNAi) suggest that general acetylation of histone H3 (or H4 elsewhere) may contribute at least as much to Brd4 recruitment, as the NSL-H4K16ac pathway. Would the authors

agree? If so, they should tone down their wording about 'a critical role'. Other HATs, such as PCAF/P300, and histone acetylation sites may be more critical than NSL.

12. Fig.6c does not illustrate the statement that "that genes whose orthologues are bound by KANSL3 in mouse ESCs had a higher probability of being downregulated in patient cells ($p= 4.8e-6$) (Fig 6c)." The lower cluster shows many up-regulated genes that are bound by Kansl3.

13. When it comes to analysis of patient-derived cells, it is said that each patient qualifies as 'biological replicate'. This is not appropriate. Biological replicates relate to the independent analysis of each patient cell line on different days, which needs to be done.

14. Whereas in *Drosophila* thousands of house-keeping genes are affected by NSL/Brd4 inhibition the number in Koolen-de Vries syndrome patients appears much smaller and indirect effects confuse the issue. The application of the suggested *Drosophila* pathway to the patient-derived fibroblasts requires looking at conserved NSL-dependent genes. The authors list just 16 genes that qualify (Fig.6d). Among these genes is Brd7, a Brd4-related protein thought to be only expressed in testes. First, the authors should show the expression levels of these 16 genes in fibroblast cells and exclude very lowly expressed ones. Lastly, the evidence for the bold statement in the title relies on less than 16 genes. In my opinion this is not sufficient to conclude that 'the crosstalk between the NSL complex and Brd4 shapes the gene expression signature in Koolen-deVries syndrome patients. The manuscript would be stronger if the *Drosophila* pathway was emphasized and the conservation of the pathway in humans would be stated as hypothetical.

Reviewer #2 (Remarks to the Author):

In this work, Gaub and colleagues identified a novel correlation between the *Drosophila* acetyltransferase complex NSL and the bromodomain containing protein dBRD4. The authors found that NSL's subunits and dBRD4 extensively co-localize on promoters in fly and mouse embryonic stem cells, and that BRD4 is required for NSL-mediated transcription and NSL complex-mediated acetylation leads to dBRD4 recruitment to chromatin. Overall the manuscript addresses an important topic that should be of interest to a broad scientific community. The cell biology data and ChIP analysis are convincing, however the NSL-dBRD4 correlation observed in cells needs to be tested/ confirmed biochemically. It is essential to find out whether dBRD4 indeed binds to H4K16ac, a mark produced by NSL, as the cell data presented in this study suggest. If dBRD4 does not recognize H4K16ac and instead binds to H4K5ac/K8ac, a possible correlation between these PTMs should be tested, especially because the authors cite the study showing that MOF is capable of acetylating H4K5/K8.

A few minor points:

- abstract- the sentence "BRD4 is required..." is unclear
- page 4, please introduce NSL3 as a subunit of the NSL complex to avoid confusion
- Is dBRD4 a common nomenclature for Fs(1)h? if not, please clarify your abbreviation
- page 13, the sentence "In addition, bulk level of H3K27ac..." is unclear and might not be of help

Point by point response to reviewers

Manuscript NCOMMS-19-1124927-T

We thank both Reviewers for their constructive comments and suggestions. We have now incorporated all the suggestions, which have significantly improved the manuscript upon revision. A point by point response to each of the comments is provided below.

We also suggest a new title for the manuscript:

New title: “Evolutionary conserved NSL complex/BRD4 axis controls transcription activation via histone acetylation.”

Previous title: “A crosstalk between the NSL complex and BRD4 shapes the gene expression signature of the Koolen-de Vries syndrome”

Reviewer #1

The manuscript by Gaub et al. describes a wealth of technically advanced experiments documenting a functional relationship between the histone acetyltransferase complex NSL and the acetyl-binding protein Brd4. The topic of the paper is of considerable interest since it deals with a novel pathway through which site-specific acetylation affect transcription.

We thank the reviewer for acknowledging the novelty of the pathway we have discovered and considering our findings of “considerable interest”. We also thank the referee for his/her very constructive experimental suggestions and careful data inspection.

Data obtained largely from the Drosophila model suggest a hierarchical scenario, according to which the H4K16 acetylation placed by NSL is bound by Brd4. Polymerase occupancy data suggest that early steps of transcription elongation may be promoted by Brd4, in keeping with the literature. Since all mentioned proteins are conserved in mammals, the authors apply the insights derived from the Drosophila model to human patients, in which the human NSL complex is impaired.

In some places the authors overstate their case. It seems that in Drosophila, NSL-mediated acetylation is one of several pathways to recruit Brd4. For the human data the situation is more complicated. I am not convinced that the manuscript keeps what is promised in the title. Below, I list observations mainly in chronological order of perceived shortcomings that should be addressed or clarified in a revised manuscript.

1. Fig.1: The screen is done well. Fig.1f may list Brd4 along with the complexes, for reference.

We thank the reviewer for this suggestion. dBRD4 has now been added to Fig. 1f.

2. Fig.2c: These polytene chromosome stainings are not informative due to lack of resolution and can be omitted.

While we agree that the polytene chromosome stainings do not match the resolution of ChIP-seq data, we prefer to keep the IF panel for a visual illustration of the NSL3 and dBRD4 co-occupancy, as this nicely complements the ChIP-seq analysis, which we also have in the manuscript.

3. Fig.2b does not really illustrate the statement in the text, namely that 86% of NSL3-bound

promoters are occupied by Brd4. Perhaps a scatter-plot is better suited, where correlations of signal strengths for individual sites are put in relation?

We thank the reviewer for this helpful input. The statement of 86% co-occupancy is based on MACS2 peak calling (see methods). We have added the requested scatter plot (Fig. 2c) and also modified the main Fig. 2b: To better illustrate the high percentage of NSL3 and dBRD4 co-occupied promoters, we have now clustered the heatmap into two groups based on NSL3 binding intensities.

4. Fig.Suppl.3: Luciferase signals are appropriately normalized to a 'renilla' control. This works if the renilla signal does not fluctuate massively, or systematically. In Fig.S3a-c it is of concern that the renilla signal systematically increases in the presence of Brd4 inhibitors, so normalization distorts the picture. The heat shock promoter apparently was a bad choice. Is there an explanation? How many promoters are positively affect by BET inhibitors?

We fully agree with the reviewer that data normalization in the case of a broad regulator such as the NSL complex is an important point.

We note that Renilla did not show a systematic response in the genome-wide screen (Supplementary Fig. 1g), neither did it respond to RNAi of the NSL complex components (see Figure below, the decrease in Renilla signal likely is a consequence of lower cell numbers e.g. upon NSL1 RNAi). This indicates that the response on hsp70 is specific for dBRD4. Hence, we think that interpretation of the screen (focused on NSLs) is not confounded by the choice of this promoter.

Figure 1: Normalized Renilla Luciferase activity (using an hsp70-driven Renilla reporter) assayed following depletion of NSL complex members by RNAi. Error bars represent standard deviation of three technical replicates.

Upon BET inhibitor treatments (Fig. 3b-d, Supplementary Fig. 3a-c), we had also noticed this change in Renilla signal intensities. For exactly this reason, the Firefly and Renilla signals are plotted separately in all of these Figure panels. Unfortunately, in the legend text we had written “ratio”, while we had plotted the separate values for Firefly and Renilla. The legend text was now corrected.

Regarding the question of how many promoters are positively affected by BET inhibitors: 200 genes show a significant upregulation after 4h of JQ1 treatment, yet a GO term analysis of these genes did not indicate GO terms related to a heat shock response. However, it is possible that iBET treatments for 9 or 16 hours (as they were performed for luciferase assays) trigger a mild stress response in the cells.

While we are not sure, why this increase in Renilla happens upon BET inhibition, we do not think this is an issue, since the focus of this manuscript is on physiologically regulated dBRD4-target genes and not this ectopic luciferase construct.

5. Fig.3f: The gene expression of NSL complex-bound genes are shown. What is missing is a control of an equivalent number of NSL-unbound genes, to establish the specificity.

We thank the reviewer for this suggestion. To address this, we tested gene expression effects for an equivalent number (2288) of NSL complex-bound and expressed and NSL-unbound and expressed genes. While the NSL complex-bound genes showed significantly lower read counts after NSL1, dBRD4 or dBRD4-L RNAi treatments, the NSL-unbound genes did not show significant differences in read counts. We have now added this information (Supplementary Fig. 3f).

6. Fig.3g: The correlation of r^2 of 0.22 is not a strong correlation as stated in the text, but at best moderate (albeit significant).

We agree with the reviewer and have now implemented the suggested change. The revised text reads: “From the global RNA-seq experiments, we uncovered **a significant correlation** between gene expression changes observed upon depletion of NSL1 and inhibition of dBRD4 by JQ1 (4h) ($adj R^2 = 0.20$, Fig. 3g).” (Page 7)

7. Fig.Suppl.3h: This is a very important control experiment, which is why the lower panel needs to show that the Brd4 RNAi has indeed worked.

We would like to point out that the upper and lower panels in Supplementary Fig. 3i (previously Supplementary Fig. 3h) belong to the same experiment (indeed, the Brd4 RNAi has worked). We agree with the reviewer that it is important to verify RNAi efficiencies in every experiment and apologize for not being clear in the legends. We have modified this figure legend now.

8. Fig.3h,i: These effects should be quantified.

As requested, immunoblots of Fig 3h and 3i have been quantified and added in Supplementary Fig. 3j.

9. Fig.Suppl.4a lacks documentation that the NSL1 RNAi worked.

As explained above, (comment 7) we have always ensured efficient knock-downs, either by ChIP or by western blot analysis. Representative examples for the NSL1 RNAi efficiency are shown in Supplementary Fig. 3i (previously Supplementary Fig. 3h) as western blot and in Supplementary Fig. 4b as ChIP experiment. We have now clarified this in the figure legend. For clarity in the manuscript, we have not always shown a panel for the knock-down efficiency of each experiment, but decided to add representative Figure panels.

10. Fig.5a: The K16ac ChIP visually appears as if there was an enhanced signal in gene bodies upon NSL1 RNAi, in NSL-bound genes. Is this the case and how could it be explained?

H4K16 acetylation is catalyzed by two distinct MOF containing complexes, the NSL complex and the Male specific lethal (MSL) complex^{1,2}. In flies, the latter is only expressed in males, where MOF functions in dosage compensation and achieves X chromosome-wide H4K16ac coating. This results in an approximately 2-fold upregulated transcriptional output of the single male X to match expression in females.

Since our experiments are conducted in male S2 cells, the slight increase of H4K16 acetylation levels on gene bodies upon NSL1 RNAi, could arise from a potential crosstalk with the MSL complex. It is possible that the MSL and NSL complexes might partially compensate for the loss of each other, a phenomenon that is apparent in e.g. MSL mutants (Valsecchi and colleagues³ and data not shown). However, this issue has not

been systematically addressed and we do not have any evidence for a molecular crosstalk in our particular experimental system.

An alternative interpretation is that the apparent increase on gene bodies is of technical nature, e.g. genome-wide changes in H4K16ac levels upon NSL1 RNAi will reflect on changes in epitope availability, which could in turn affect the background levels of the IP and/or IP efficiencies.

As this increase observed in our experiments is rather minor, we prefer not to interpret this slight change in H4K16 acetylation levels here.

11. Fig.5.,Suppl.5: The effect of HDACi are mainly interpreted in the context of H4K16ac levels, but 5b (NSL1 RNAi) suggest that general acetylation of histone H3 (or H4 elsewhere) may contribute at least as much to Brd4 recruitment, as the NSL-H4K16ac pathway. Would the authors agree? If so, they should tone down their wording about 'a critical role'. Other HATs, such as PCAF/P300, and histone acetylation sites may be more critical than NSL.

We thank the reviewer for this important comment. The experiment presented in Figure 5 shows that depletion of NSL1 leads to a up to 50% reduction of dBRD4 from promoters. As we had stated in the text, this indicates that other HATs are likely to contribute to the recruitment as well, but in our view, this does not make the contribution of the NSL complex less critical.

We now strengthened this connection by performing an *in vitro* peptide binding assay. We found that dBRD4 displays strong binding to di-acetylated H4K12, K16ac and pan-acetylated H4 peptides, whereas its binding to mono acetylated H4K12ac or H4K16ac peptides is within the background range of our assay (data added in Supplementary Fig. 5a-c). Importantly, the NSL complex can catalyze acetylation at multiple Lysine positions along the H4 tail *in vitro*^{2,4}. Together these results suggest that a combinatorial effect of acetylation marks deposited by the NSL complex or other HATs mediates dBRD4 recruitment.

The complexity of HAT regulation, their essentiality and presence of feedback loops makes it difficult to address, which HAT is the most critical for the recruitment of BRD4 (also see the comment above regarding crosstalk with the MSL complex). Therefore, we have followed the recommendation of the reviewer and have ensured that the phrasing was appropriately toned down where necessary.

12. Fig.6c does not illustrate the statement that "that genes whose orthologues are bound by KANSL3 in mouse ESCs had a higher probability of being downregulated in patient cells ($p=4.8e-6$) (Fig 6c)." The lower cluster shows many up-regulated genes that are bound by Kansl3.

We thank the reviewer for raising this point. The statistics were done using directionality of gene changes (Figure 6g), whereas only significantly affected genes were plotted in the heatmap of Figure 6c. To avoid confusion, we have therefore removed the KANSL3 ChIP panel in Figure 6c.

In addition, we have changed the wording in the text to: "*that genes whose orthologues are bound by KANSL3 in mouse ESCs had a significantly higher probability of being downregulated in patient cells ($p=4.8e-6$) (Fig. 6c).*" (Page 11)

13. When it comes to analysis of patient-derived cells, it is said that each patient qualifies as 'biological replicate'. This is not appropriate. Biological replicates relate to the independent analysis of each patient cell line on different days, which needs to be done.

We would like to clarify, that for the purpose of identifying commonly misregulated genes in Koolen-de Vries patient-derived fibroblasts, we defined the different patient-derived cell lines as replicates for the bioinformatic analysis. We are aware, that this approach has disadvantages and it will not allow to capture gene expression profiles of each individual.

To address the concern, we have now added qPCR expression analyses of four biological replicates of each of the patient-derived cell lines for a subset of targets, that we had previously identified in RNA-seq (see Supplementary Fig. 6f and g). We are happy to report that these validations (now added in Supplementary Fig. 7a and b) support that our RNA-seq data is reliable.

14. Whereas in Drosophila thousands of house-keeping genes are affected by NSL/Brd4 inhibition the number in Koolen-de Vries syndrome patients appears much smaller and indirect effects confuse the issue. The application of the suggested Drosophila pathway to the patient-derived fibroblasts requires looking at conserved NSL-dependent genes. The authors list just 16 genes that qualify (Fig.6d). Among these genes is Brd7, a Brd4-related protein thought to be only expressed in testes. First, the authors should show the expression levels of these 16 genes in fibroblast cells and exclude very lowly expressed ones. Lastly, the evidence for the bold statement in the title relies on less than 16 genes. In my opinion this is not sufficient to conclude that ‘the crosstalk between the NSL complex and Brd4 shapes the gene expression signature in Koolen-deVries syndrome patients. The manuscript would be stronger if the Drosophila pathway was emphasized and the conservation of the pathway in humans would be stated as hypothetical.

We thank the reviewer for raising this concern. We would like to point out that we did have a threshold to exclude very lowly expressed genes from the RNAseq analysis (baseMean < 10).

The expression levels of the 16 conserved genes in the fibroblast RNAseq dataset are listed in the table below. The baseMean is given, which represents the average normalized count value across all samples. The baseMean count of a few housekeeping and pluripotency genes are listed as a control in a separate table. BRD7 (ENSG00000166164) is ubiquitously expressed (<https://www.proteinatlas.org/ENSG00000166164-BRD7/tissue>).

Conserved Gene List:

Gene	baseMean
TMEM165	4279
SCP2	3847
MTMR6	2043
NUP58	1721
ATG3	1464
BRD7	1404
MICU2	1141
UTP11	1120
DMAC2	1078
MRPL11	690
SNIP1	618
DIAPH2	584
EXOSC3	461
NACAD	394
RBL1	361
MED30	191
ABHD3	113

Reference Example Gene List:

Gene	baseMean
GAPDH	96688
HPRT1	568
KAT8 (MOF)	586
KANSL1	1055
NANOG	0
POU5F1	1
ESRRB	1

We agree that the mammalian data (mouse ES cells and human patient-derived cells) presented in our manuscript does not have the mechanistic breath of the *Drosophila* experiments. Nevertheless, we are convinced that our study provides clear indications that the NSL-BRD4 crosstalk is conserved in mammals. Collectively, our findings provide an important perspective towards the understanding of the metabolic character of the Koolen-de Vries syndrome and the *Drosophila* and mammalian data together provide a strong conceptual novelty regarding NSL/BRD4 regulating cellular homeostasis genes.

We have now ensured that our manuscript is appropriately toned down at the respective passages in the text. Our new title now emphasizes the conservation aspect of the *Drosophila* pathway, instead of the Koolen-de Vries syndrome:

“Evolutionary conserved NSL complex/BRD4 axis controls transcription activation via histone acetylation.”

Reviewer #2

In this work, Gaub and colleagues identified a novel correlation between the Drosophila acetyltransferase complex NSL and the bromodomain containing protein dBRD4. The authors found that NSL's subunits and dBRD4 extensively co-localize on promoters in fly and mouse embryonic stem cells, and that BRD4 is required for NSL-mediated transcription and NSL complex-mediated acetylation leads to dBRD4 recruitment to chromatin. Overall the manuscript addresses an important topic that should be of interest to a broad scientific community.

We thank the referee for highlighting the importance of our work for a broad scientific community. We would particularly like to thank the referee for his/her very constructive experimental input. This helped us to further strengthen the molecular mechanism of the interplay between the NSL complex, the Histone H4 tail and BRD4.

The cell biology data and ChIP analysis are convincing, however the NSL-dBRD4 correlation observed in cells needs to be tested/confirmed biochemically. It is essential to find out whether dBRD4 indeed binds to H4K16ac, a mark produced by NSL, as the cell data presented in this study suggest. If dBRD4 does not recognize H4K16ac and instead binds to H4K5ac/K8ac, a possible correlation between these PTMs should be tested, especially because the authors cite the study showing that MOF is capable of acetylating H4K5/K8.

We thank the reviewer for raising this point and are happy to report that we could now characterize the dBRD4-H4K16ac interaction in biophysical assays (now added in Supplementary Fig. 5a-c).

Previously, binding affinities for bromodomains of the human BRD4 proteins to differentially modified histone peptides had been tested, demonstrating a cooperative binding mode of the bromodomains to multiple acetylated histone residues⁵.

We successfully achieved the stable expression of the full length, recombinant 110 kDa dBRD4 protein in bacteria. This allowed us to perform biolayer interferometry experiments for measuring the binding strength of dBRD4 to N-terminal histone H4 peptides harboring different acetylations. An important advantage of using biolayer interferometry is that interactions between a ligand and analyte can be directly quantified from lysates (<https://www.moleculardevices.com/en/assets/app-note/biologics/instant-determination-of-protein-presence-using-blitz-system>,⁶)(note, that it was not possible to

establish purification of the full-length protein; the protein appeared stable in the extract, but underwent rapid aggregation once it was purified, i.e. when it was taken out of the “context” of the extract. Apparently, the full-length protein requires a “cellular” environment).

We observed that indeed H4K16ac promotes binding of dBRD4 to acetylated histone peptides. Interestingly, similar to previous reports, dBRD4 expressing lysates also showed cooperative binding when multiple sites were acetylated on histone H4 tail peptide (see Supplementary Fig. 5a-c for details). An important control in these experiments are HP1^{SWI6} expressing lysates, which did not display binding to acetylated histone H4 peptides, while they did show robust interaction with H3K9me3. These *in vitro* results demonstrate that histone H4K16 acetylation can be recognized by dBRD4 and thus can contribute to the recruitment of dBRD4 to chromatin.

A few minor points:

- abstract- the sentence “BRD4 is required...” is unclear

As requested, we have rephrased this sentence in the abstract. It now reads as: “*Using Drosophila and mouse embryonic stem cells, we unravel a recruitment hierarchy, where NSL-deposited histone acetylation induces BRD4 recruitment for transcription of constitutively active genes.*”

- page 4, please introduce NSL3 as a subunit of the NSL complex to avoid confusion

We have added this information, the sentence now reads as: “*To this end, we performed a genome-wide RNAi screen based on a luciferase reporter assay in cultured cells, where tethering of NSL3, a subunit of the NSL complex, to a minimal promoter through a Gal4-DNA binding domain conveys strong transcriptional activity (Fig. 1a, Supplementary Fig. 1a).*” (Page 4)

- Is dBRD4 a common nomenclature for Fs(1)h? if not, please clarify your abbreviation

The abbreviation dBRD4 for Fs(1)h has been previously used in the following publication: Kang *et al.* 2017 ⁷. Other authors have used Brd4 as abbreviation for Fs(1)h ⁸, however to avoid confusion with the mouse protein, we have here decided to use dBRD4 for the *Drosophila* Fs(1)h protein.

- page 13, the sentence “In addition, bulk level of H3K27ac...” is unclear and might not be of help

We thank the reviewer for helping us to refine the text. This paragraph has been modified, it reads now:

“Our data show that the NSL complex contributes to chromatin recruitment of BRD4 in Drosophila and mice through acetylation of the histone 4 tail. This is in agreement with a recent small-scale study in human cells showing that depletion of MOF and subsequent reduction in H4K16ac correlate with loss of BRD4 from promoters of several autophagy-related genes ⁹. Nevertheless, since combinatorial effects of multiple acetylated histone residues enhance the affinity of BET proteins to histones (Supplementary Fig. 5 and ^{5,10,11}), it is likely that additional acetyl marks on the histone octamer, mediated by other histone acetyl transferases, such as P300, may also impact on chromatin recruitment of BRD4 and dBRD4.” (Page 14)

References:

1. Akhtar, A. & Becker, P.B. Activation of transcription through histone H4 acetylation by MOF, an acetyltransferase essential for dosage compensation in *Drosophila*. *Molecular cell* **5**, 367-375 (2000).
2. Cai, Y. et al. Subunit composition and substrate specificity of a MOF-containing histone acetyltransferase distinct from the male-specific lethal (MSL) complex. *Journal of Biological Chemistry* **285**, 4268-4272 (2010).
3. Valsecchi, C.I.K. et al. Facultative dosage compensation of developmental genes on autosomes in *Drosophila* and mouse embryonic stem cells. *Nature communications* **9**, 3626 (2018).
4. Chatterjee, A. et al. MOF acetyl transferase regulates transcription and respiration in mitochondria. *Cell* **167**, 722-738. e23 (2016).
5. Filippakopoulos, P. et al. Histone recognition and large-scale structural analysis of the human bromodomain family. *Cell* **149**, 214-231 (2012).
6. Sultana, A. & Lee, J.E. Measuring protein-protein and protein-nucleic acid interactions by biolayer interferometry. *Current protocols in protein science* **79**, 19.25. 1-19.25. 26 (2015).
7. Kang, H. et al. Bivalent complexes of PRC1 with orthologs of BRD4 and MOZ/MORF target developmental genes in *Drosophila*. *Genes & development* **31**, 1988-2002 (2017).
8. Haberle, V. et al. Transcriptional cofactors display specificity for distinct types of core promoters. *Nature* **570**, 122 (2019).
9. Sakamaki, J.-i. et al. Bromodomain protein BRD4 is a transcriptional repressor of autophagy and lysosomal function. *Molecular Cell* **66**, 517-532. e9 (2017).
10. Dey, A., Chitsaz, F., Abbasi, A., Misteli, T. & Ozato, K. The double bromodomain protein Brd4 binds to acetylated chromatin during interphase and mitosis. *Proceedings of the National Academy of Sciences* **100**, 8758-8763 (2003).
11. Morinière, J. et al. Cooperative binding of two acetylation marks on a histone tail by a single bromodomain. *Nature* **461**, 664 (2009).

REVIEWERS' COMMENTS:

Reviewer #1 (Remarks to the Author):

The authors addressed the issues I raised to my satisfaction. They toned down some of the statements and found a more appropriate title. The revised manuscript can be published.

Reviewer #2 (Remarks to the Author):

The authors have adequately addressed my previous comments, best regards, Tatiana Kutateladze

Re: NCOMMS-19-1124927A
RESPONSE TO REVIEWER'S COMMENTS

Reviewer #1

The authors addressed the issues I raised to my satisfaction. They toned down some of the statements and found a more appropriate title. The revised manuscript can be published.

We thank the reviewer for her/his support and critical advice which helped to significantly improve our manuscript.

Reviewer #2

The authors have adequately addressed my previous comments, best regards, Tatiana Kutateladze

We thank the reviewer for her support and critical advice which helped to significantly improve our manuscript.